# An advantage based policy transfer algorithm for reinforcement learning with measures of transferability

## Abstract

Reinforcement learning (RL) enables sequential decision-making in complex and high-dimensional environments through interaction with the environment. In most real-world applications, however, a high number of interactions are infeasible. In these environments, transfer RL algorithms, which can be used for the transfer of knowledge from one or multiple source environments to a target environment, have been shown to increase learning speed and improve initial and asymptotic performance. However, most existing transfer RL algorithms are on-policy and sample inefficient, fail in adversarial target tasks, and often require heuristic choices in algorithm design. This paper proposes an off-policy Advantage-based Policy Transfer algorithm, APT-RL, for fixed domain environments. Its novelty is in using the popular notion of "advantage" as a regularizer, to weigh the knowledge that should be transferred from the source, relative to new knowledge learned in the target, removing the need for heuristic choices. Further, we propose a new transfer performance measure to evaluate the performance of our algorithm and unify existing transfer RL frameworks. Finally, we present a scalable, theoretically-backed task similarity measurement algorithm to illustrate the alignments between our proposed transferability measure and similarities between source and target environments. We compare APT-RL with several baselines, including existing transfer-RL algorithms, in three high-dimensional continuous control tasks. Our experiments demonstrate that APT-RL outperforms existing transfer RL algorithms and is at least as good as learning from scratch in adversarial tasks.

## 1 Introduction

Real-world implementation of reinforcement learning (RL) often utilizes transfer learning as a practical approach to solve sequential decision-making problems data efficiently. The concept of transfer learning in the context of RL denotes the transfer of knowledge from one or multiple source environments to a target environment (Kaspar et al., 2020; Zhao et al., 2020; Bousmalis et al., 2018; Peng et al., 2018; Yu et al., 2017). Formally, this problem setting can be explained using two Markov decision processes (MDP), target MDP $\mathcal{M}_\mathcal{T} = \langle \mathcal{X}, \mathcal{A}, \mathcal{R}_\mathcal{T}, \mathcal{P}_\mathcal{T} \rangle$ and source MDP $\mathcal{M}_\mathcal{S} = \langle \mathcal{X}, \mathcal{A}, \mathcal{R}_\mathcal{S}, \mathcal{P}_\mathcal{S} \rangle$ where $\mathcal{X}$ is the state-space, $\mathcal{A}$ is the action-space, $\mathcal{R}_\mathcal{T}$ and $\mathcal{R}_\mathcal{S}$ are the target and source reward functions, $\mathcal{P}_\mathcal{T}$ and $\mathcal{P}_\mathcal{S}$ are the target and source transition dynamics. The source environment, $\mathcal{M}_\mathcal{S}$, can be a simulated or physical environment which, if learned, provides some sort of useful knowledge to be transferred to $\mathcal{M}_\mathcal{T}$. While in general $\mathcal{X}$ and $\mathcal{A}$ could be different between the source and target environments, we consider *fixed domain* environments, where a domain is defined as $\langle \mathcal{X}, \mathcal{A} \rangle$ and is assumed to be identical in the source and target.

For effective transfer of knowledge between fixed domain environments, one would expect $\mathcal{R}_\mathcal{S}$ and $\mathcal{P}_\mathcal{S}$ to be similar $\mathcal{R}_\mathcal{T}$ and $\mathcal{P}_\mathcal{T}$, by either intuition, heuristics, or by some codified metric. The similarity between these MDP components may determine the effectiveness of knowledge transfer between the source and target. To illustrate this, let us consider the single source transfer application to the four-room toy example (Fig. 1). The objective is to learn a target policy, $\pi_\mathcal{T}^*$, by utilizing knowledge from the source task. To demonstrate the effect of target and source similarity, we consider two different target tasks that are slightly different in terms of the dynamics and the goal in Fig. 1(b) and (c). The objective is to learn the optimal policies $\pi_{\mathcal{T}_1}^*$ and $\pi_{\mathcal{T}_2}^*$ in the corresponding target environments, $\mathcal{T}_1$ and $\mathcal{T}_2$, by utilizing knowledge from the source task, $\mathcal{S}$. Intuitively, $\mathcal{T}_1$

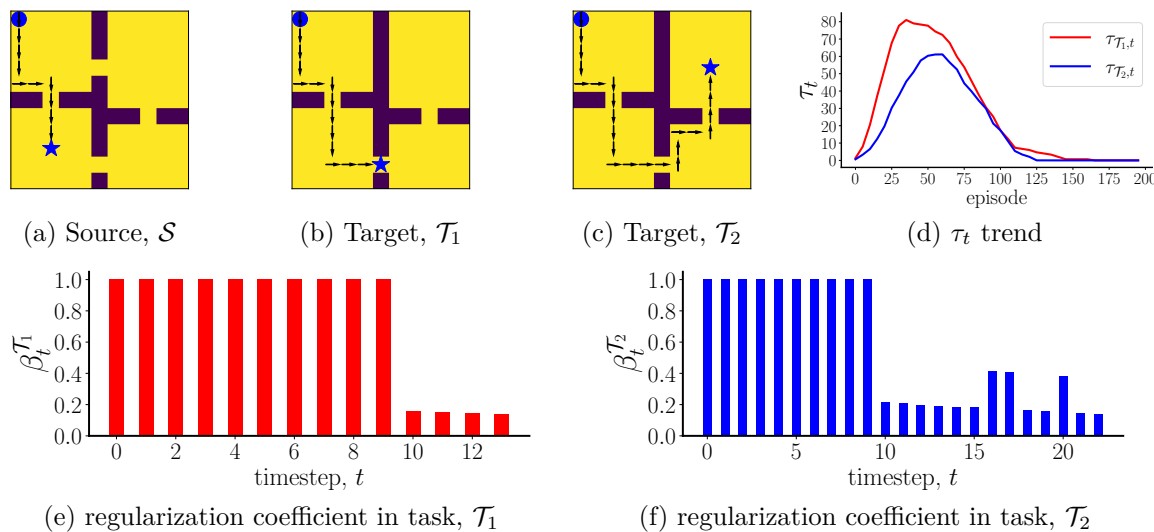

Figure 1: Knowledge transfer in the four-room toy problem. (a)–(c) show the source task $\mathcal{S}$ and two target tasks $\mathcal{T}_1, \mathcal{T}_2$ (●: start, ★: goal). (d) shows the evaluation measure $\tau_t$ in the two target tasks. (e)–(f) show the influence of the source policy on each target, plotted as regularization coefficient $\beta_t$. Note that the source can influence target $\mathcal{T}_1$ more than target $\mathcal{T}_2$ which matches the intuition of (d)

is more similar to the source task $\mathcal{S}$ when compared to $\mathcal{T}_2$. As a result, we expect that knowledge transferred from $\mathcal{S}$ to $\mathcal{T}_1$ will be comparatively more useful. $\mathcal{T}_2$ is less similar to $\mathcal{S}$ and thus knowledge transferred from $\mathcal{S}$ to $\mathcal{T}_2$ may be useful, but should be less effective than that transferred to $\mathcal{T}_1$. Based on this motivating example, we can identify three fundamental challenges in developing efficient transfer RL algorithms; 1) developing measure to quantify the effectiveness of knowledge transfer, 2) calculating the task similarity, and finally 3) developing practical algorithms for effective knowledge transfer. In this study, we propose new ways to address these major challenges.

To quantify the knowledge transfer in RL setting, different measures have been introduced in literature but there is a lack of unified approach that can help us compare between multiple different transfer strategies. To this end, one of our first contributions is the introduction of the notion of *transferability evaluation measure*, $\tau_t$ (section 4.1). We propose transferability evaluation measure as a function of evaluation episode $t$ and can be considered as a generalized concept that can express different evaluation measure of transfer learning such as sample complexity, reward accumulation etc. As an example, for the Four-room toy problem, Figure 1(d) shows that the transferability evaluation measure $\tau_{\mathcal{T}_2,t}$ for $\mathcal{T}_2$ is useful ($\tau_{\mathcal{T}_2,t} > 0$), but not as useful as the transfer to $\mathcal{T}_1$ ($\tau_{\mathcal{T}_1,t} > \tau_{\mathcal{T}_2,t}$). This insight also supports the natural connection between transferability and task similarity. Thus, we argue that task similarity measurement approach can provide critical insights about the usefulness of the source knowledge, and as we show, can also be leveraged in comparing the performance of different transfer RL algorithms. To illustrate this further, we propose a theoretically-backed, model-based simple *task similarity measurement* algorithm, and use it to show that our proposed transferability measure closely aligns with the similarities between source and target tasks (section 4.2.2).

Finally, we propose a new transfer RL algorithm that uses the popular notion of "advantage" as a regularizer, to weigh the knowledge that should be transferred from the source, relative to new knowledge learned in the target. The main benefit of our proposed transfer RL algorithm is to utilize the source policy while learning the target policy without relying on manual hyperparameter. To achieve this, first we calculate the advantage function based on actions selected using the source policy. Next, we use this advantage function as a regularization coefficient to weigh the influence of the source policy on the target. As a motivating example, we show the effectiveness of this idea in our earlier toy example in terms of regularization weight[1], $\beta$. Figure 1(f) shows that $\beta_t^{\mathcal{T}_1}$ is lower than $\beta_t^{\mathcal{T}_2}$. This result intuitively means that $\mathcal{S}$ can provide useful guidance for

---

[1] we use the exponential of the advantage function as the regularization weight, $\beta = \exp A(\mathbf{x}, \mathbf{a})$

most of the actions selected by the target policy except the last four actions in $\mathcal{T}_1$, whereas, in contrast, the guidance is poor for the last thirteen actions in $\mathcal{T}_2$. We show that this simple yet scalable framework can improve transfer learning capabilities in RL. Our proposed advantage-based policy transfer algorithm is straightforward to implement and, as we show empirically on several continuous control benchmark gym environments, can be at least as good as learning from scratch in adversarial source tasks.

Our main contributions are the following:

- We propose a novel advantage-based policy transfer algorithm, APT-RL, that performs better than previous policy transfer algorithms from literature for fixed-domain tasks in RL

- We propose a new *relative transfer performance* measure to evaluate and compare the performance of transfer RL algorithms. Our idea extends the previously proposed formal definition of transfer in RL by Lazaric (2012), and unifies previous approaches proposed by (Taylor et al., 2007; Zhu et al., 2020). We provide theoretical support for the effectiveness of this measure (Theorem 1) and demonstrate its use in the evaluation of APT-RL on different benchmarks and against other algorithms (Section 6).

- We propose a simple model-based *task similarity measurement* algorithm, and use it to illustrate the relationship between source and target task similarities and our proposed transferability measure (Section 6). We motivate this algorithm by providing new theoretical bounds on the difference in the optimal policies' action-value functions between the source and target environments in terms of gaps in their environment dynamics and rewards (Theorem 2).

- We demonstrate the performance of APT-RL in twelve high-dimensional continuous control tasks. In terms of data efficiency, our algorithm performs better than zero-shot source policy transfer, SAC (Haarnoja et al., 2018) without any source knowledge, fine-tuning the source policy with target data, and REPAINT, a state-of-the-art transfer RL algorithm (Tao et al., 2020).

The paper is organized as follows: we begin by reviewing related work in section 2, We propose and explain our off-policy transfer RL algorithm, APT-RL, in section 3. Next, we propose evaluation measures for APT-RL and a scalable task similarity measurement algorithm between fixed domain environments in section 4. We outline our experiment setup in section 5, and use the proposed transferability metric and task similarity measurement algorithm to evaluate the performance of APT-RL and compare it against other algorithms in section 6.

## 2 Related work

Traditionally, transfer in RL is described as the transfer of knowledge from one or multiple source tasks to a target task to help the agent learn in the target task. Transfer achieves one of the following: a) increases the learning speed in the target task, b) jumpstarts initial performance, and/or c) improves asymptotic performance (Taylor & Stone, 2009). Transfer in RL has been studied extensively in the literature, as evidenced by the three surveys (Taylor & Stone, 2009; Lazaric, 2012; Zhu et al., 2020) on the topic.

The prior literature proposes several approaches for transferring knowledge in RL. One such approach is the transfer of instances or samples (Lazaric et al., 2008; Taylor et al., 2008). Another approach is learning some representation from the source and then transferring it to the target task (Taylor & Stone, 2007b; 2005). Sometimes the transfer is also used in RL for better generalization between several environments instead of focusing on sample efficiency. For example, Barreto et al. (2017); Zhang et al. (2017) used successor feature representation to decouple the reward and dynamics model for better generalization across several tasks with similar dynamics but different reward functions. Another approach considered policy transfer where the KL divergence between point-wise local target trajectory deviation is minimized and an additional intrinsic reward is calculated at each timestep to adapt to a new similar task (Joshi & Chowdhary, 2021). In contrast to these works, our proposed approach simply uses the notion of advantage function to transfer *policy parameters* to take knowledge from the source policy to the target task and thus the transfer of knowledge is automated without depending on any heuristic choice.

There have also been different approaches to comparing source and target tasks and evaluating task similarity. A few studies have focused on identifying similarities between MDPs in terms of state similarities or policy similarities (Ferns et al., 2004; Castro, 2020; Agarwal et al., 2021). A couple of studies also focused on transfer RL where each of the MDP elements is varying (Taylor & Stone, 2007a; Gupta et al., 2017). Most of these approaches either require a heuristic mapping or consider a high level of similarity between the source and target tasks. In contrast to these works, we develop a scalable task similarity measurement algorithm for fixed domain environments that does not require the learning of the optimal policy.

A few recent studies have focused on fixed domain transfer RL problems for high-dimensional control tasks (Zhang et al., 2018; Tao et al., 2020). Most of these studies are built upon on-policy algorithms which require online data collection and tend to be less data efficient than an off-policy algorithm (which we consider here). Although Zhang et al. (2018) discussed off-policy algorithms briefly along with decoupled reward and transition dynamics, a formal framework is absent. Additionally, learning decoupled dynamics and reward models accurately is highly non-trivial and requires a multitude of efforts. More recently, Tao et al. (2020) proposed an on-policy actor-critic framework that utilizes the source policy and off-policy instance transfer for learning a target policy. This idea is similar to our approach, but different in two main ways. First, we consider an entirely off-policy algorithm, unlike Tao et al. (2020), and second, our approach does not require a manually tuned hyperparameter for regularization. Additionally, Tao et al. (2020) discards samples collected from the target environment that do not follow a certain threshold value which hampers data efficiency. Finally, Tao et al. (2020) only considers environments where source and target only vary by rewards and not dynamics. In contrast, we account for varying dynamics, which we believe to be of practical importance for transfer RL applications.

In addition to transfer RL, several recent studies have proposed algorithms for cross-domain policy adaptation. In particular, DARC (Eysenbach et al., 2020), VGDF (Xu et al., 2023), IGDF (Wen et al., 2024), and PAR (Lyu et al., 2024) proposes algorithms that collect source data during target policy update for adaptation. In contrast, we study transfer RL setting where we assume no access to the source environment once the knowledge is transferred to the target.

## 3 APT-RL: An off-policy advantage based policy transfer algorithm

In this section, we present our proposed transfer RL algorithm. We explain two main ideas that are novel to this algorithm: advantage-based regularization, and synchronous updates of the source policy. As our proposed algorithm notably utilizes advantage estimates to control the weight of the policy from a source task, we call it **A**dvantage based **P**olicy **T**ransfer in **RL** (APT-RL). We build upon soft-actor-critic (SAC) (Haarnoja et al., 2018), a state-of-the-art off-policy RL algorithm for model-free learning. We use off-policy learning to re-use past experiences and make learning sample-efficient. In contrast, an on-policy algorithm would require collecting new samples for each gradient update, which often makes the number of samples required for learning considerably high. Our choice of off-policy learning therefore helps with the scalability of APT-RL to higher dimensional problems, which we will show in the case studies.

### 3.1 Advantage-based policy regularization

The first new idea in our algorithm is to consider utilizing source knowledge during each gradient step of the policy update. Our intuition is that the current policy, $\pi_\phi$ parameterized by $\phi$, should be close to the source optimal policy, $\pi^*_{\mathcal{S}}$, when the source can provide useful knowledge. In contrast, when the source knowledge does not aid learning in the target, then less weight should be put on the source knowledge. Based on this intuition, we modify the policy update formula of SAC as follows: we use an additional regularization loss with a temperature parameter, along with the original SAC policy update loss, to control the effect of the added regularization loss. Formally, the policy parameter has the update formula:

$$\phi \leftarrow \phi + \delta_\pi \left[ \hat{\nabla}_\phi J_1(\phi) + \beta_t \hat{\nabla}_\phi J_2(\phi) \right] \tag{1}$$

Pwhere $J_1(\cdot)$ is the usual SAC policy update loss, which uses soft-Q values instead of Q-values, and Q-function is parameterized by $\theta$ for the dataset $\mathcal{D}_{\mathcal{T}}$, and entropy regularization coefficient $\alpha$ and learning rate

$\delta_\pi$,

$$J_1(\phi) = \mathbb{E}_{\mathbf{x}_t \sim \mathcal{D}_\mathcal{T}} \left[ \mathbb{E}_{\mathbf{a}_t \sim \pi_\phi} [\alpha \log(\pi_\phi(\mathbf{a}_t|\mathbf{x}_t)) - Q_\theta(\mathbf{x}_t, \mathbf{a}_t)] \right], \tag{2}$$

and $J_2(\cdot)$ is the cross-entropy loss, $\mathcal{H}(\cdot, \cdot)$, between the source policy and the current policy,

$$J_2(\phi) = \mathcal{H}(\mu_\psi(\mathbf{a}|\mathbf{x}), \pi_\phi(\mathbf{a}|\mathbf{x})) = \mathbb{E}_{\mu_\psi(\mathbf{a}|\mathbf{x})}[-\log \pi_\phi(\mathbf{a}|\mathbf{x})], \tag{3}$$

where $\mu_\psi$ represents the optimal source policy $\pi_\mathcal{S}^*$ parameterized by $\psi$.

Thus, we are biasing the current policy $\pi_\phi$ to stay close to the source optimal policy $\pi_\mathcal{S}^*$ by minimizing the cross-entropy between these two policies while using the temperature parameter $\beta$ to control the effect of the source policy. Typically, this type of temperature parameter is considered a hyper-parameter and requires (manual) fine-tuning. Finding an appropriate value for this parameter is highly non-trivial and maybe even task-specific. Additionally, if the value of $\beta$ is not appropriately chosen, then the effect of the source policy may be detrimental to learning in the target task.

To overcome these limitations, we propose an *advantage-based* control of the temperature parameter, $\beta$. The main motivation here is to make the source influence adaptive, which means that we do not treat $\beta$ as a hyper-parameter. The core intuition of our idea is that the second term of Equation 1 should have more weight when the average action taken according to $\mu_\psi$ is better than the random action. If the source policy provides an action that is worse than a random action in the target, then $\mu_\psi$ should be regularized to have less weight. This is equivalent to taking the difference of the advantages based on the current policy and the source policy, respectively. In addition, we consider the exponential, rather than the absolute value, of this difference, so that the temperature approaches zero when the source provides adversarial knowledge. Formally, our proposed advantage-based temperature parameter is determined as follows:

$$\beta_t = e^{A_\mathcal{S}^t - A_\mathcal{T}^t}, A_\mathcal{T}^t = Q_\theta(\mathbf{x}_t, \pi_\phi(\mathbf{x}_t)) - V(\mathbf{x}_t), A_\mathcal{S}^t = Q_\theta(\mathbf{x}_t, \mu_\psi(\mathbf{x}_t)) - V(\mathbf{x}_t) \tag{4}$$

Note that we can leverage the relationship between the soft-Q values and soft-value functions to represent the advantages from Equation 4 in a more convenient way that follows from (Haarnoja et al., 2018):

$$\begin{aligned}
A_\mathcal{T}^t &= Q_\theta(\mathbf{x}_t, \pi_\phi(\mathbf{x}_t)) - \mathbb{E}[Q_\theta(\mathbf{x}_t, \mu_\psi(\mathbf{x}_t)) - \alpha \log \mu_\psi(\mathbf{a}_t|\mathbf{x}_t)] \\
A_\mathcal{S}^t &= Q_\theta(\mathbf{x}_t, \pi_\phi(\mathbf{x}_t)) - \mathbb{E}[Q_\theta(\mathbf{x}_t, \pi_\phi(\mathbf{x}_t)) - \alpha \log \pi_\phi(\mathbf{a}_t|\mathbf{x}_t)]
\end{aligned} \tag{5}$$

## 3.2 Synchronous update of the source policy

We propose an additional improvement over the advantage-based policy transfer idea to further improve sample efficiency. As we consider a parameterized source optimal policy, $\mu_\psi$, it is possible to update the parameters of the source policy with the target data, $\mathcal{D}_\mathcal{T}$, by minimizing the SAC loss. The benefits of this approach is two-fold: 1) If the source optimal policy provides useful information to the target, then this will accelerate the policy optimization procedure by working as a regularization term in Equation 1, and 2) This approach enables sample transfer to the target policy. This is because the initial source policy is learned using the source data; when this source policy is further updated with the target data, it can be viewed as augmenting the source dataset with the latest target data. Formally, the source policy will be updated as follows: $\psi \leftarrow \psi + \delta_\psi \hat{\nabla}_\psi J_1(\psi)$, where $J_1(\psi)$ is the typical SAC loss for the source policy with parameters $\psi$ and learning rate $\delta_\psi$. The pseudo-code for APT-RL is shown in Algorithm 1.

## 4 An evaluation framework for transfer RL

To formally quantify the performance of APT-RL, including against other algorithms, in this section, we propose notions of transferability and task similarity, as discussed in Section 1. First, we propose a formal notion of transferability and use this notion to calculate a "relative transfer performance" measure. We demonstrate how this measure can be utilized to assess and compare the performance of APT-RL and similar algorithms. Then, we propose a scalable task similarity measurement algorithm for high-dimensional environments. We motivate this algorithm by providing a new theoretical bound on the difference in the optimal policies action-value functions between the source and target environments in terms of gaps in their

---

**Algorithm 1** APT-RL: **A**dvantage based **P**olicy **T**ransfer in **R**einforcement **L**earning

---

1: **Given:** parameterized source optimal policy $\mu_\psi$, source learning rate $\delta_\mathcal{S}$, target learning rate $\delta_\pi$
2: **Initialize:** current target policy, $\pi_\phi$, target buffer $\mathcal{D}_\mathcal{T} = \emptyset$
3: **for** each iteration **do**
4:     **for** each target environment step **do**
5:         $\mathbf{a} \sim \pi_\phi(\mathbf{a}_t | \mathbf{x}_t)$
6:         $\mathbf{x}' \sim p(\mathbf{x}' | \mathbf{x}, \mathbf{a})$
7:         $\mathcal{D}_\mathcal{T} \leftarrow \mathcal{D}_\mathcal{T} \cup \{(\mathbf{x}, \mathbf{a}, \mathcal{R}(\mathbf{x}, \mathbf{a}), \mathbf{x}')\}$
8:     **end for**
9:     **for** $G$ gradient updates **do**
10:         $\psi \leftarrow \psi + \delta_\mathcal{S} \hat{\nabla}_\psi J_1(\psi)$  where $J_1$ is defined in Equation 2
11:         calculate $A_\mathcal{T}^t$ and $A_\mathcal{S}^t$ using Equation 5
12:         $\beta_t \leftarrow e^{A_\mathcal{S}^t - A_\mathcal{T}^t}$
13:         $\phi \leftarrow \phi + \delta_\pi \left[ \hat{\nabla}_\phi J_1(\phi) + \beta_t \hat{\nabla}_\phi J_2(\phi) \right]$
14:     **end for**
15: **end for**

---

environment dynamics and rewards. This task similarity measurement algorithm can be used to identify the best source task for transfer. Further, we use this algorithm in the experiment section, to illustrate how our proposed relative transfer performance closely aligns with the similarities between the source and target environment.

### 4.1   A measure of transferability

We formally define *transferability* as a mapping from stationary source knowledge and non-stationary target knowledge accumulated until timestep $t$, to learning performance, $\rho_t$.

**Definition 4.1** (Single-task transferability)**.** Let $\mathcal{K}_\mathcal{S}$ be the transferred knowledge from a source task $\mathcal{M}_\mathcal{S}$ to a target task $\mathcal{M}_\mathcal{T}$, and let $\mathcal{K}_{\mathcal{T},t}$ be the available knowledge in $\mathcal{M}_\mathcal{T}$ at timestep $t$. Let $\rho_t \in \mathbb{R}$ denote a measure that evaluates the learning performance in $\mathcal{M}_\mathcal{T}$ at timestep $t$. Then, transferability is defined as the mapping,

$$\Lambda : \mathcal{K}_\mathcal{S} \times \mathcal{K}_{\mathcal{T},t} \to \mathbb{R} \ .$$

Intuitively, this means that $\Lambda(\cdot)$ takes prior source knowledge and accumulated target knowledge to evaluate the learning performance in the target task.

As an example, if the collection of source data samples $\mathcal{D}_\mathcal{S}$ are utilized as the transferred knowledge and the average returns using the latest target policy, $G_t = \mathbb{E}^{\pi_{\mathcal{T},t}} \left[ \sum_{k=0}^\infty \gamma^k r_k | \mathbf{x}_0 \right]$, is used as the evaluation performance of a certain transfer algorithm $i$, then the transferability of algorithm $i$, $\Lambda_i$, at episode $t$, can be represented using $\mathcal{D}_\mathcal{S}$ as $\mathcal{K}_\mathcal{S}$, $\mathcal{D}_{\mathcal{T},t}$ as $\mathcal{K}_{\mathcal{T},t}$, and finally $G_t$ as $\rho_t$.

| source knowledge, $\mathcal{K}_\mathcal{S}$ | target knowledge, $\mathcal{K}_{\mathcal{T},t}$ | evaluation measure, $\rho_t$ |
|---|---|---|
| samples, $\mathcal{D}_\mathcal{S}$ | samples, $\mathcal{D}_{\mathcal{T},t}$ | average returns, $G_t = \mathbb{E}^{\pi_{\mathcal{T},t}} \left[ \sum_{k=0}^H r_k \right]$ |
| policy, $\mu_\psi$ | current policy, $\pi_{\mathcal{T},t}$ | samples required for reward threshold, $n_{\text{th}}$ |
| models $\mathcal{R}_\mathcal{S}, \mathcal{P}_\mathcal{S}$ | models, $\mathcal{R}_{\mathcal{T},t}, \mathcal{P}_{\mathcal{T},t}$ | area under the reward curve, $\Delta_t$ |
| value functions $Q_\mathcal{S}^*, V_\mathcal{S}^*$ | value functions $Q_{\mathcal{T},t}, V_{\mathcal{T},t}$ | samples required for asymptotic returns, $n_\infty$ |

Table 1: A list of potential source knowledge, target knowledge and evaluation performance measures. Target knowledge and evaluation performance are represented for episode $t$, and $\pi_{\mathcal{T},t}$ denotes the optimal target policy at evaluation episode $t$.

Notice that we leave the choice of input and output of this mapping as user-defined task-specific parameters. Any traditional transfer methods can be represented using the idea of transferability. Potential choices for source and target knowledge and evaluation measures are listed in Table 1. Our idea extends the previously

proposed formal definition of transfer in RL by Lazaric (2012), and unifies previous approaches proposed by Taylor et al. (2007); Zhu et al. (2020). Expressing transfer learning algorithms in terms of this notion of transferability has a number of advantages. First, this problem formulation can be easily extended to unify other important RL settings. For example, this definition can be extended to offline RL (Levine et al., 2020) by considering $\mathcal{K}_{\mathcal{S}} = \mathcal{D}_{\text{source}}$, and $\mathcal{K}_{\mathcal{T},t} = \emptyset$, $\forall t$. Second, the comparison of two transfer methods becomes convenient if they have the same evaluation criteria. For instance, one way to construct evaluation criteria may be to use sample complexity in the target task to achieve a desired return. Subsequently, the transferability measure can be used to assess "relative transfer performance", which can act as a tool for comparing two different transfer methods conveniently.

**Definition 4.2** (Relative transfer performance, $\tau$). Given the transferability mapping of algorithm $i$, $\Lambda_i$, the relative transfer performance is defined as the difference between the corresponding learning performance $\rho_t^i$ and learning performance from a base RL algorithm $\rho_t^b$ at evaluation episode $t$. Formally, $\tau_t = \rho_t^i - \rho_t^b$ , where the base RL algorithm represents learning from scratch in the target task (meaning $\mathcal{K}_{\mathcal{S}} = \emptyset$), and $\rho_t^i$, $\rho_t^b$ are the same evaluation criteria of learning performances.

### 4.1.1 Theoretical support

We first formally show that, with an appropriate definition of the evaluation measure, non-negative relative transfer performance leads to a policy in the target task which is at least as good as learning from scratch.

**Theorem 1. (Relative transfer performance and policy improvement)** *Consider $\rho_t^i = \mathbb{E}^{\pi_{i,t}}\left[\sum_{k=0}^{\infty} \gamma^k r_k | \mathbf{x}_0\right]$ for policy $\pi_i$ and $\rho_t^b = \mathbb{E}^{\pi_{b,t}}\left[\sum_{k=0}^{\infty} \gamma^k r_k | \mathbf{x}_0\right]$ for policy $\pi_b$, where $\mathbf{x}_0$ is the starting state and each policy is executed until termination condition. Then, the learned policy, $\pi_{i,t}$ using algorithm $i$, in the target at episode $t$ is at least as good as the source optimal policy $\pi_{b,t}$ if $\tau_t \geq 0$.*

The proof can be found in the Appendix A.

### 4.1.2 Revisiting the toy problem

We also leverage the toy example presented in Fig. 1 to explain the proposed concepts. The performance evaluation is chosen as the average returns, collected from an evaluation episode $t$, that is $\rho_t = \sum_{k=0}^{H} r_k$. For transfer, we choose the low-level direct knowledge Q-values. At first, we initialize the target Q-values with pre-trained source Q-values, $Q_{\mathcal{S}}^*$. Thus, at each episode, $t$, the updated Q-values in the target are a combination of both source and target knowledge. Thus, the transferability mapping can be expressed as $\Lambda_{\text{Q-learning}} : \mathcal{K}_{\mathcal{S}} \times \mathcal{K}_{\mathcal{T},t} \to \mathbb{R}$. We calculate $\rho_t$ after every 10 timestep by executing the greedy policy from the updated Q-values for a fixed time horizon. Relative transfer performance, $\tau_t$, remains non-negative for both the target tasks $\mathcal{T}_1$ and $\mathcal{T}_2$, for up to around 125 evaluation episodes. Intuitively this means that the learning performance of both policies is better than a base algorithm for all of the evaluation episodes. Also, $\tau_t$ is higher for $\mathcal{T}_1$ than $\mathcal{T}_2$ which means that transferring knowledge from $\mathcal{S}$ leads to better learning performance in $\mathcal{T}_1$ compared to $\mathcal{T}_2$. This can be explained by the fact that the dynamics in $\mathcal{T}_1$ are more similar to $\mathcal{S}$ than $\mathcal{T}_2$.

## 4.2 Measuring task similarity

As seen in the toy problem above, a measure of task similarity would help us illustrate the close alignment of our proposed transferability metric with the similarities of the source and target tasks. Beyond this, measuring task similarity can provide additional insights into why a particular source task is more appropriate to transfer knowledge to the target task. Motivated by these, we propose an algorithm for measuring task similarity in this section.

### 4.2.1 Theoretical motivation

To motivate the idea behind our proposed algorithm, we first investigate theoretical bounds on the expected discrepancies between the policies learned in the source and target environments. One effective way for this analysis is to calculate the upper bound on differences between the optimal policies' action-values. Previously, action-value bounds have been proposed for similar problems by Csáji & Monostori (2008); Abdolshah et al.

(2021). The simulation lemma (Kearns & Singh, 2002; Jiang, 2020) also shows the value-function bound between two MDPs for any policy where one MDP is a sufficient approximation of another MDP. Motivated by previous literature, we demonstrate the action-value bound between target MDP with target optimal policy and target MDP with source optimal policy.

We extend these ideas to the transfer learning setting where we derive the bound between the target action-values under target optimal policy, $\pi_{\mathcal{T}}^*$ and target action-values under source optimal policy, $\pi_{\mathcal{S}}^*$.

**Theorem 2.** *(Action-value bound between fixed-domain environments) If $\pi_{\mathcal{S}}^*$ and $\pi_{\mathcal{T}}^*$ are the optimal policies in the MDPs $\mathcal{M}_{\mathcal{S}} = \langle \mathcal{X}, \mathcal{A}, \mathcal{R}_{\mathcal{S}}, \mathcal{P}_{\mathcal{S}} \rangle$ and $\mathcal{M}_{\mathcal{T}} = \langle \mathcal{X}, \mathcal{A}, \mathcal{R}_{\mathcal{T}}, \mathcal{P}_{\mathcal{T}} \rangle$ respectively, then the corresponding action-value functions is upper bounded by*

$$||\mathbf{Q}_{\mathcal{T}}^{\pi_{\mathcal{T}}^*} - \mathbf{Q}_{\mathcal{T}}^{\pi_{\mathcal{S}}^*}||_\infty \le \frac{2\delta_{\mathcal{ST}}^r}{1-\gamma} + \frac{2\gamma\delta_{\mathcal{ST}}^{TV}(R_{max,\mathcal{S}} + R_{max,\mathcal{T}})}{(1-\gamma)^2} \tag{6}$$

*where $\delta_{\mathcal{ST}}^r = ||\mathcal{R}_{\mathcal{S}}(\mathbf{x},\mathbf{a}) - \mathcal{R}_{\mathcal{T}}(\mathbf{x},\mathbf{a}))||_\infty$, $\delta_{\mathcal{ST}}^{TV}$ is the total variation distance between $\mathcal{P}_{\mathcal{S}}$ and $\mathcal{P}_{\mathcal{T}}$, $\gamma$ is the discount factor and $R_{max,\mathcal{S}} = ||\mathcal{R}_{\mathcal{S}}(\mathbf{x},\mathbf{a})||_\infty$, $R_{max,\mathcal{T}} = ||\mathcal{R}_{\mathcal{T}}(\mathbf{x},\mathbf{a})||_\infty$.*

The proof of this theorem can be found in the Appendix B. Note that as we propose APT-RL for fixed-domain environments, the bound can be expressed in terms of differences in the remaining environment parameters: the total variation distance between the source and target transition probabilities, and the maximum reward difference between the source and the target. Intuitively this bound means that a lower total variation distance between the transition dynamics can provide a tighter bound on the deviation between the action-values from the target and source optimal policies. Similarly, having a smaller reward difference also helps in getting lower action-value deviations in the target. Also note that, if the reward function or transition dynamics remain identical between source and target, then the corresponding term on the right side of Equation 6 vanishes.

Next, motivated by this bound, we propose a task similarity measurement algorithm that assesses the differences between source and target dynamics and rewards in order to evaluate their similarity.

### 4.2.2 A model-based task similarity measurement algorithm

Previous attempts for measuring task similarity include behavioral similarities in MDP in terms of state-similarity or bisimulation metric, and policy similarity (Ferns et al., 2004; Castro, 2020; Agarwal et al., 2021). Calculating such metrics in practice is often challenging due to scalability issues and computation limits. Additionally, our key motivation is to find a similarity measurement that does not require solving for the optimal policy *apriori*, as the latter is often the key challenge in RL. To this end, we propose a new model-based method for calculating similarities between tasks.

We propose an encoder-decoder based deep neural network model at the core of this idea. For any source or target task, a dataset, $\mathcal{D} = \{(\mathbf{x}, \mathbf{a}, r, \mathbf{x}')\}$, is collected by executing a random policy. Next, a dynamics model, $f_i^{\mathcal{P}}(\mathbf{x}, \mathbf{a})$, for task $i$ is trained by minimizing the mean-squared-error (MSE) loss using stochastic gradient descent, $\mathcal{L}_{\text{dyn}} = ||\mathbf{x}' - f_i^{\mathcal{P}}(\mathbf{x}, \mathbf{a})||_2$. Similarly, a reward model, $f^{\mathcal{R}}(\mathbf{x}, \mathbf{a})$ is trained using the collected data to minimize the following MSE loss, $\mathcal{L}_{\text{rew}} = ||r - f_i^{\mathcal{R}}(\mathbf{x}, \mathbf{a})||_2$. The encoder portion of the neural network model encodes state and action inputs into a latent vector. Then, the decoder portion uses this latent vector for the prediction of the next state or reward. We consider decoupled models for this purpose, meaning that we learn separate models for reward and transition dynamics from the same dataset. This allows us to identify whether only reward or transition dynamics or both vary between tasks. Once these models are learned, the source model is used to predict the target data and calculate the $L_2$ distance between the predicted and actual data as the similarity error. If $\xi_k^{\mathcal{P}}$ and $\xi_k^{\mathcal{R}}$ are the similarity errors in target dynamics and rewards, respectively, we can calculate the dynamics and reward similarity separately as follows,

$$\text{dynamics similarity: } \Xi_{\mathcal{S},\mathcal{T}}^{\mathcal{P}} = \frac{1}{|\mathcal{D}_{\mathcal{T}}|}\sum_{k=1}^{|\mathcal{D}_{\mathcal{T}}|} \xi_k^{\mathcal{P}}, \text{reward similarity: } \Xi_{\mathcal{S},\mathcal{T}}^{\mathcal{R}} = \frac{1}{|\mathcal{D}_{\mathcal{T}}|}\sum_{k=1}^{|\mathcal{D}_{\mathcal{T}}|} \xi_k^{\mathcal{R}} \tag{7}$$

Our approach is summarized in Algorithm 2. This approach can be viewed as a modern version of (Ammar et al., 2014), but instead of using Restricted Boltzmann Machines, we use deep neural network-based encoder-decoder architecture to learn the models, and we do it in a decoupled way.

---

**Algorithm 2** Model-based task similarity measurement

---

1: Collect $m$ data samples from source $\mathcal{M}_{\mathcal{S}}$ using a random policy, $\mathcal{D}_{\mathcal{S}} = \{(\mathbf{x}_{\mathcal{S}}, \mathbf{a}_{\mathcal{S}}, r_{\mathcal{S}}, \mathbf{x}'_{\mathcal{S}})\}$
2: Collect $m$ data samples from target $\mathcal{M}_{\mathcal{T}}$ using a random policy, $\mathcal{D}_{\mathcal{T}} = \{(\mathbf{x}_{\mathcal{T}}, \mathbf{a}_{\mathcal{T}}, r_{\mathcal{T}}, \mathbf{x}'_{\mathcal{T}})\}$
3: Learn target dynamics and reward models by minimizing $||\mathbf{x}' - f_{\mathcal{T}}^{\mathcal{P}}(\mathbf{x}, \mathbf{a})||_2$, $||r - f_{\mathcal{T}}^{\mathcal{R}}(\mathbf{x}, \mathbf{a})||_2$ using $\mathcal{D}_{\mathcal{T}}$
4: Learn source dynamics and reward models by minimizing $||\mathbf{x}' - f_{\mathcal{S}}^{\mathcal{P}}(\mathbf{x}, \mathbf{a})||_2$, $||r - f_{\mathcal{S}}^{\mathcal{R}}(\mathbf{x}, \mathbf{a})||_2$ using $\mathcal{D}_{\mathcal{S}}$
5: **for** each $(\mathbf{x}_{\mathcal{T}}, \mathbf{a}_{\mathcal{T}}, r_{\mathcal{T}}, \mathbf{x}'_{\mathcal{T}}) \in \mathcal{D}_{\mathcal{T}}$ **do**
6: $\quad \hat{\mathbf{x}}'_{\mathcal{T}} = f_{\mathcal{T}}^{\mathcal{P}}(\mathbf{x}_{\mathcal{T}}, \mathbf{a}_{\mathcal{T}}), \hat{r}_{\mathcal{T}} = f_{\mathcal{T}}^{\mathcal{R}}(\mathbf{x}_{\mathcal{T}}, \mathbf{a}_{\mathcal{T}})$
7: $\quad \xi_k^{\mathcal{P}} = ||\hat{\mathbf{x}}'_{\mathcal{T}} - \mathbf{x}'_{\mathcal{T}}||_2, \xi_k^{\mathcal{R}} = ||\hat{r}_{\mathcal{T}} - r_{\mathcal{T}}||_2$
8: **end for**
9: dynamics similarity, $\Xi_{\mathcal{S},\mathcal{T}}^{\mathcal{P}} = \frac{1}{|\mathcal{D}^{\mathcal{T}}|} \sum_{k=1}^{|\mathcal{D}^{\mathcal{T}}|} \xi_k^{\mathcal{P}}$
10: reward similarity, $\Xi_{\mathcal{S},\mathcal{T}}^{\mathcal{R}} = \frac{1}{|\mathcal{D}^{\mathcal{T}}|} \sum_{k=1}^{|\mathcal{D}^{\mathcal{T}}|} \xi_k^{\mathcal{R}}$

---

## 5 Experiment Setup

We apply the described methods to three popular high-dimensional continuous control benchmark gym environments (Brockman et al., 2016): 1) 'HalfCheetah-v3', 2) 'Ant-v3', and 3) 'Humanoid-v3'. The vanilla Gym environments are not suitable for transfer learning settings. We change the original parameters of the task to create four different perturbations of the dynamics of each environment. For the HalfCheetah-v3 environment, we consider the original task as the source with standard gym values for damping while the four target environments have different values of damping increased gradually in each task. As a result, the least similar task has the highest damping values in the joints. For the Ant-v3 environment, the source task has the standard gym robot and the four target environments have varied dynamics by changing the leg lengths of the robot. Similarly, for the Humanoid-v3 environment, the source task has the standard gym robot and the four target environments have varied dynamics by changing the leg length as well as the size of the hands, legs and shin length. For the HalfCheetah-v3 environment, we create additional four tasks by changing the reward function to show reward variation. Details of the environments and the algorithm hyperparameters can be found in Appendices C and D.

For each environment, the knowledge transferred is the source optimal policy $\pi_{\mathcal{S}}^*$, data collected from the target task $\mathcal{D}_{\mathcal{T},t}$ is used as the target knowledge, and average returns during each evaluation episode, $\mathbb{E}^{\pi_{\mathcal{T},t}}[\sum_k r_k]$, are used as the performance evaluation metric. To obtain the source optimal policy, the SAC algorithm is utilized to train a policy from scratch in each source environment. In each of these experiments, we perform and compare our algorithm APT-RL against one of the recent benchmarks on transfer RL, the REPAINT algorithm proposed by Tao et al. (2020). We also compare APT-RL against zero-shot source policy transfer, policy fine-tuning on the target task, and SAC without any source knowledge.

## 6 Experiment Results

### 6.1 Task similarity

We leverage Algorithm 2 to calculate the task similarity between the source and each of the target tasks. The empirical task similarity is shown in Fig. 2. Fig. 2(a) shows the similarity in tasks for the half-cheetah environment with varying rewards. For reward similarity, we can see the highest dissimilarity when the robot is provided a negative reward instead of a positive reward. Similarly, for the half-cheetah environment with varying dynamics in Fig. 2(b), we can see an approximately linear trend in the similarity of the dynamics. This makes sense as the change in joint damping is gradual and constant. In Fig. 2(c) we show the task similarity in the Ant environment with varying dynamics. As we change the dynamics of each of the target environments by changing the length of the legs of the robot, the similarity between each target and the

source task reduces monotonically. Finally, task similarity of the humanoid environments are shown in Fig. 2(d) where first two tasks are relatively more similar to the source and the final two tasks are less similar.

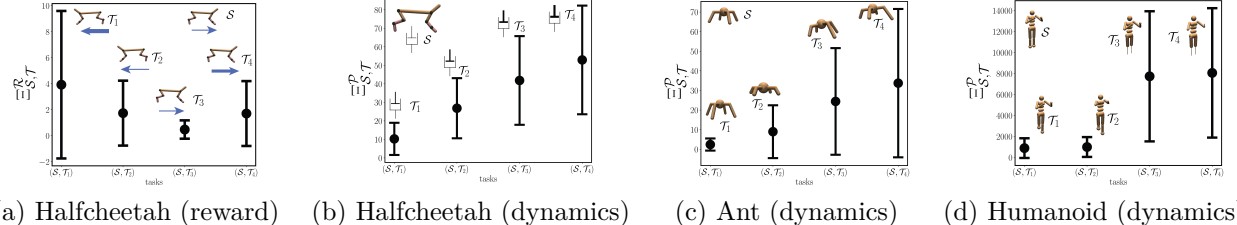

| (a) Halfcheetah (reward) | (b) Halfcheetah (dynamics) | (c) Ant (dynamics) | (d) Humanoid (dynamics) |

Figure 2: **Task dissimilarity:** Empirical task similarity between several variations of Half-cheetah, Ant, and Humanoid environments

.

## 6.2 Transferability of APT-RL, $\Lambda_{\textsf{APT-RL}}$

### 6.2.1 Half-cheetah-v3

Fig. 3 (a)-(c) shows the transfer evaluation performance for three target tasks with varying dynamics and Fig. 3(d) shows the performance for one target task with negative reward. In all cases, APT-RL learns faster and also achieves higher average returns than learning from scratch. While fine-tuning the source policy performs better than APT-RL in task $\mathcal{T}_1$, it performs worse than both APT-RL and learning from scratch in rest of the tasks in $\mathcal{T}_2, \mathcal{T}_3, \mathcal{T}_4$. Most importantly, APT-RL performs as good as learning from scratch for target task $\mathcal{T}_4$ with a negative reward. Note that, this is an adversarial source task as the robot needs to learn to run in the opposite direction in the target task. In contrast, the REPAINT algorithm fails to achieve similar evaluation performance and in most cases, obtains evaluation performance lower than learning from scratch using SAC. As REPAINT is a PPO-based on-policy algorithm, this result aligns with the previously reported performance of SAC and PPO algorithms (Haarnoja et al., 2018). The performance of APT-RL may be explained by the fact that the source optimal policy jumpstarts the target policy. This increase in learning performance, in turn, provides a positive relative transfer measure over time in tasks $\mathcal{T}_1, \mathcal{T}_2, \mathcal{T}_3$ as shown in Fig. 4. Note that $\tau$ remains higher for the most similar task $\mathcal{T}_1$ and relatively lower for the least similar task $\mathcal{T}_2$. Temperature parameter $\beta$ decreases quickly initially, then increases slightly and stays approximately constant over time. This can be explained by the fact that more weight is put into the regularization loss initially and once the target policy becomes better the effect reduces. As we keep utilizing the target data to update the source policy, the source policy improves over time and provides useful information during the later timesteps. Finally, we observe that the effect of the source policy remains almost constant with respect to task similarity. This makes sense due to the particular change in dynamics of the environment. We anticipate that changing only the damping values make the tasks less adversarial to the source in terms of dynamics.

### 6.2.2 Ant-v3

For the ant environment, we observe significant performance gain of APT-RL in the target task against learning from scratch, zero-shot policy transfer, fine-tuning, and the REPAINT algorithm (Fig. 3). For target tasks that are very similar to the source, we observe a fast convergence of the policy in the target. For less similar source and target, APT-RL can even achieve higher learning performance than learning from scratch. This might happen due to the jumpstart of the target policy and also due to the synchronous improvement of the source policy. The latter characteristic of APT-RL accelerates policy updates. Similar to the half-cheetah environment, we observe that the temperature parameter decreases with timesteps and decreases more when task similarity is lower (Fig. 4). In all of these examples, APT-RL outperforms the REPAINT algorithm.

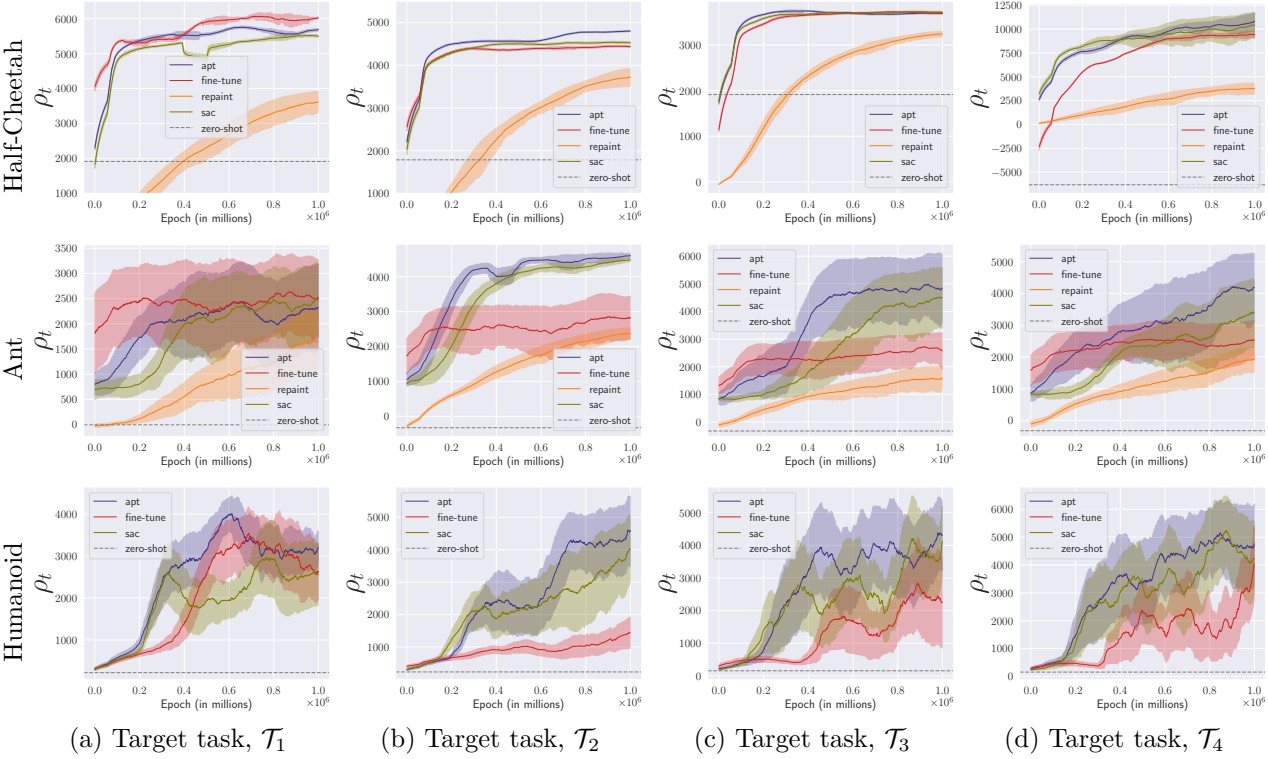

(a) Target task, $\mathcal{T}_1$     (b) Target task, $\mathcal{T}_2$     (c) Target task, $\mathcal{T}_3$     (d) Target task, $\mathcal{T}_4$

Figure 3: **APT-RL transferability, $\Lambda_{\mathbf{APT\text{-}RL}}$:** APT-RL is compared against vanilla SAC (learning from scratch), REPAINT, zero-shot policy, and fine-tuned policy. Average return during the evaluation episode is taken as $\rho_t$, meaning $\rho_t = \mathbb{E}^{\pi^*_{\mathcal{T}_i}}[\sum_t r_k]$. We do not show Repaint for the humanoid environment as it fails to solve the tasks. Results are shown with one standard deviation range.

### 6.3 Humanoid-v3

For the humanoid environment, APT-RL outperforms all the baselines with significant initial performance gain (Fig. 3). For the final target task, $\mathcal{T}_4$, APT-RL performs similarly to learning from scratch. We anticipate that this behavior is mainly due to the difficulty of the task. The change in the dynamics of the environment make the target an adversarial task which is relatively more difficult to solve than the rest of the tasks. This can be also supported by the fact that $\mathcal{T}_4$ is the least similar task among all of the target tasks. Similar to the ant and half-cheetah environments, we observe that the temperature parameter decreases with less task similarity (Fig. 4). We do not show results from the REPAINT algorithm as it fails to solve even the source task.

Finally, we show an ablation study on the effect of the temperature parameter, $\beta$, in Fig. 5. Notice that APT-RL outperforms manual choice of $\beta$ in all tasks except the halfcheetah environment where $\beta = 1.0$ performs slightly better than APT-RL. Interestingly, Fig. 4 shows that APT-RL also converges to $\beta = 1.0$ without any manual choice. We argue that it shows further evidence on the strength of APT-RL where we do not need to manually choose the temperature parameter.

## 7 Limitations

The task similarity algorithm presented in section 4.2.2 uses the random policy to learn models for the dynamics and the reward. While using a random policy for model learning is fairly common in the literature Zhang et al. (2018); Moerland et al. (2023), a key challenge is to learn a reasonable model for complex tasks. We anticipate that the task similarity algorithm might not be effective in environments where a random policy cannot efficiently capture the underlying complexities of the dynamics and the reward. We would

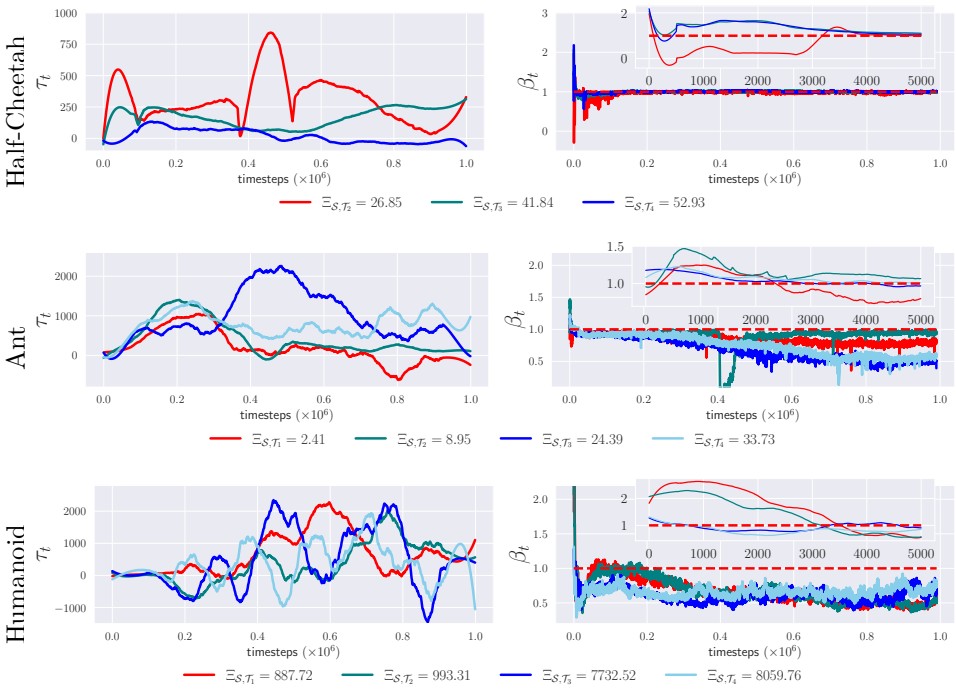

Figure 4: Left: Relative transfer performance, $\tau_t$ are shown with corresponding mean similarity scores. Right: Regularization co-efficient, $\beta_t$, is shown for all tasks with corresponding mean similarity scores.

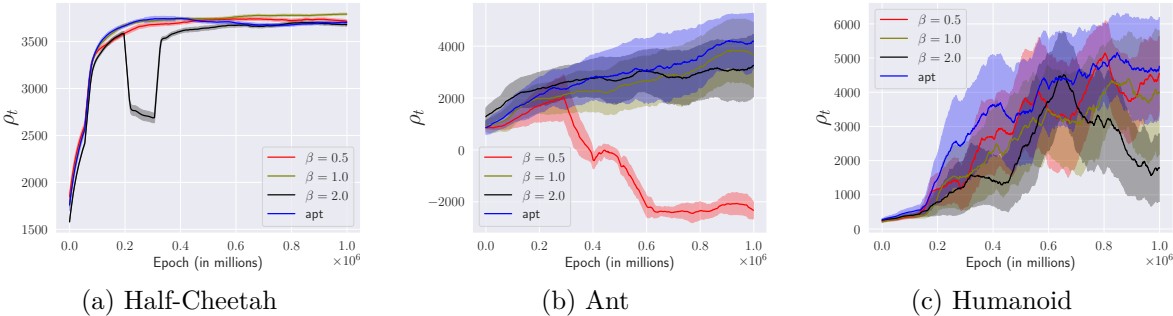

(a) Half-Cheetah          (b) Ant          (c) Humanoid

Figure 5: Ablation study of $\beta$ parameter in APT-RL: manual tuning of hyperparameter $\beta$ is shown against APT-RL in the least similar tasks for all three environments.

also like to focus on the limitations of APT-RL. One of the key challenges of transfer RL is to identify useful source tasks and reduce the impact of adversarial sources. While APT-RL shows strong performance in most of the cases, as shown in the Fig. 3(a) for the 'Half-Cheetah-V3' environment, simple transfer approaches such as fine-tuning the source policy is more convenient. This behavior can be explained by the close similarity of the source to the target. Therefore, identifying measures of when to opt for more advanced transfer algorithms such as APT-RL remains an interesting challenge. Additionally, we anticipate that APT-RL might be less effective in knowledge transfer where the source and optimal policies differ greatly due to the policy regularization.

## 8 Conclusion

In this paper, we proposed the APT-RL algorithm to transfer knowledge from a source task in an off-policy fashion. Through advantage-based regularization, our algorithm does not require any heuristic or manual

fine-tuning of the objective function. We also introduced a new relative transfer performance measure, which can help evaluate and compare transfer learning approaches in RL. We also provided a simple, theoretically-backed algorithm to calculate task similarity, and demonstrated the alignment of our proposed transfer performance measure with source and target task similarities. We demonstrated the effectiveness of APT-RL in continuous control tasks and showed its superior performance against benchmark transfer RL algorithms. Future directions may include considering similar concepts for multi-task transfer learning scenarios, as well as benchmarking the performance of various transfer learning algorithms with the help of the transferability measures introduced in this paper.

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

## A  Proof of Theorem 1

**Theorem 1.** *(Relative transfer performance and policy improvement) Consider $\rho_t^i = \mathbb{E}^{\pi_{i,t}}\left[\sum_{k=0}^{\infty}\gamma^k r_k|\mathbf{x}_0\right]$ for policy $\pi_i$ and $\rho_t^b = \mathbb{E}^{\pi_{b,t}}\left[\sum_{k=0}^{\infty}\gamma^k r_k|\mathbf{x}_0\right]$ for policy $\pi_b$, where $\mathbf{x}_0$ is the starting state and each policy is executed until termination condition, then the learned policy, $\pi_{i,t}$ using algorithm $i$, in the target at episode $t$ is at least as good as the source optimal policy, $\pi_{b,t}$ if $\tau_t \geq 0$.*

*Proof.* Let us consider $\tau_t = \mathbb{E}^{\pi_{i,t}}[G|\mathbf{x}_0] - \mathbb{E}^{\pi_{b,t}}[G|\mathbf{x}_0]$ where $\pi_i$ is the current policy in the task $i$ and $\pi_b$ is the optimal base policy and $\mathbf{x}_0$ is the initial state drawn from the same distribution. Using this definition of $\tau$, we can write the following for any starting state $s_0$:

$$\tau_t \geq 0$$
$$\Rightarrow \mathbb{E}^{\pi_{i,t}}[G|\mathbf{x}_0] - \mathbb{E}^{\pi_{b,t}}[G|\mathbf{x}_0] \geq 0$$
$$\Rightarrow V_{\pi_{i,t}}(\mathbf{x}_0) - V_{\pi_{b,t}}(\mathbf{x}_0) \geq 0$$
$$\Rightarrow V_{\pi_{i,t}}(\mathbf{x}_0) \geq V_{\pi_{b,t}}(\mathbf{x}_0)$$

Following the policy improvement theorem, we can say that $\pi_{i,t}$ is at least as good as $\pi_{b,t}$. $\square$

## B  Proof of Theorem 2

**Theorem 2.** *(Action-value bound between fixed-domain environments) If $\pi_{\mathcal{S}}^*$ and $\pi_{\mathcal{T}}^*$ are the optimal policies in the MDPs $\mathcal{M}_{\mathcal{S}} = \langle \mathcal{X}, \mathcal{A}, \mathcal{R}_{\mathcal{S}}, \mathcal{P}_{\mathcal{S}}\rangle$ and $\mathcal{M}_{\mathcal{T}} = \langle \mathcal{X}, \mathcal{A}, \mathcal{R}_{\mathcal{T}}, \mathcal{P}_{\mathcal{T}}\rangle$ respectively, then the corresponding action-value functions can be upper bounded by*

$$||\mathbf{Q}_{\mathcal{T}}^{\pi_{\mathcal{T}}^*} - \mathbf{Q}_{\mathcal{T}}^{\pi_{\mathcal{S}}^*}||_{\infty} \leq \frac{2\delta_{\mathcal{S}\mathcal{T}}^r}{1-\gamma} + \frac{2\gamma\delta_{\mathcal{S}\mathcal{T}}^{TV}(R_{max,\mathcal{S}} + R_{max,\mathcal{T}})}{(1-\gamma)^2} \tag{8}$$

*where $\delta_{\mathcal{S}\mathcal{T}}^r = ||\mathcal{R}_{\mathcal{S}}(\mathbf{x},\mathbf{a}) - \mathcal{R}_{\mathcal{T}}(\mathbf{x},\mathbf{a}))||_{\infty}$, $\delta_{\mathcal{S}\mathcal{T}}^{TV}$ is the total variation distance between $\mathcal{P}_{\mathcal{S}}$ and $\mathcal{P}_{\mathcal{T}}$, $\gamma$ is the discount factor and $R_{max,\mathcal{S}} = ||\mathcal{R}_{\mathcal{S}}(\mathbf{x},\mathbf{a})||_{\infty}$, $R_{max,\mathcal{T}} = ||\mathcal{R}_{\mathcal{T}}(\mathbf{x},\mathbf{a})||_{\infty}$.*

*Proof.* Let us consider the following notations for simplicity, $Q_i^{\pi_i^*}(\mathbf{x},\mathbf{a}) \equiv Q_i^i(\mathbf{x},\mathbf{a}), Q_j^{\pi_j^*}(\mathbf{x},\mathbf{a}) \equiv Q_j^j(\mathbf{x},\mathbf{a}), Q_i^{\pi_j^*}(\mathbf{x},\mathbf{a}) \equiv Q_i^j(\mathbf{x},\mathbf{a})$. Now we can write the following,

$$|Q_i^i(\mathbf{x},\mathbf{a}) - Q_i^j(\mathbf{x},\mathbf{a})| = |Q_i^i(\mathbf{x},\mathbf{a}) - Q_j^j(\mathbf{x},\mathbf{a}) + Q_j^j(\mathbf{x},\mathbf{a}) - Q_i^j(\mathbf{x},\mathbf{a})|$$
$$\leq \underbrace{|Q_i^i(\mathbf{x},\mathbf{a}) - Q_j^j(\mathbf{x},\mathbf{a})|}_{(a)} + \underbrace{|Q_j^j(\mathbf{x},\mathbf{a}) - Q_i^j(\mathbf{x},\mathbf{a})|}_{(b)}$$

Our strategy is to find bounds for (a) and (b) separately and then combine them to get the final bound.

**(a)**

$$|Q_i^i(\mathbf{x}, \mathbf{a}) - Q_j^j(\mathbf{x}, \mathbf{a})|$$

$$= \left| r_i(\mathbf{x}, \mathbf{a}) + \gamma \sum_{\mathbf{x}'} p_i(\mathbf{x}'|\mathbf{x}, \mathbf{a}) \max_{\mathbf{b}} Q_i^i(\mathbf{x}', \mathbf{b}) - r_j(\mathbf{x}, \mathbf{a}) - \gamma \sum_{\mathbf{x}'} p_j(\mathbf{x}'|\mathbf{x}, \mathbf{a}) \max_{\mathbf{b}} Q_j^j(\mathbf{x}', \mathbf{b}) \right|$$

$$\leq |r_i(\mathbf{x}, \mathbf{a}) - r_j(\mathbf{x}, \mathbf{a})| + \gamma \left| \sum_{\mathbf{x}'} \underbrace{p_i(\mathbf{x}'|\mathbf{x}, \mathbf{a})}_{p_i} \max_{\mathbf{b}} Q_i^i(\mathbf{x}', \mathbf{b}) - \sum_{\mathbf{x}'} \underbrace{p_j(\mathbf{x}'|\mathbf{x}, \mathbf{a})}_{p_j} \max_{\mathbf{b}} Q_j^j(\mathbf{x}', \mathbf{b}) \right|$$

$$= |r_i(\mathbf{x}, \mathbf{a}) - r_j(\mathbf{x}, \mathbf{a})| + \gamma \left| \sum_{\mathbf{x}'} p_i \max_{\mathbf{b}} Q_i^i(\mathbf{x}', \mathbf{b}) - p_j \max_{\mathbf{b}} Q_i^i(\mathbf{x}', \mathbf{b}) + p_j \max_{\mathbf{b}} Q_i^i(\mathbf{x}', \mathbf{b}) - p_j \max_{\mathbf{b}} Q_j^j(\mathbf{x}', \mathbf{b}) \right|$$

$$= |r_i(\mathbf{x}, \mathbf{a}) - r_j(\mathbf{x}, \mathbf{a})| + \gamma \left| \sum_{\mathbf{x}'} (p_i - p_j) \max_{\mathbf{b}} Q_i^i(\mathbf{x}', \mathbf{b}) + p_j \left( \max_{\mathbf{b}} Q_i^i(\mathbf{x}', \mathbf{b}) - \max_{\mathbf{b}} Q_j^j(\mathbf{x}', \mathbf{b}) \right) \right|$$

$$\leq |r_i(\mathbf{x}, \mathbf{a}) - r_j(\mathbf{x}, \mathbf{a})| + \gamma \sum_{\mathbf{x}'} \left| (p_i - p_j) \max_{\mathbf{b}} Q_i^i(\mathbf{x}', \mathbf{b}) + p_j \left( \max_{\mathbf{b}} Q_i^i(\mathbf{x}', \mathbf{b}) - \max_{\mathbf{b}} Q_j^j(\mathbf{x}', \mathbf{b}) \right) \right|$$

$$\leq \underbrace{|r_i(\mathbf{x}, \mathbf{a}) - r_j(\mathbf{x}, \mathbf{a})|}_{T_1} + \underbrace{\gamma \sum_{\mathbf{x}'} \left| (p_i - p_j) \max_{\mathbf{b}} Q_i^i(\mathbf{x}', \mathbf{b}) \right|}_{T_2} + \underbrace{\gamma \sum_{\mathbf{x}'} \left| p_j \left( \max_{\mathbf{b}} Q_i^i(\mathbf{x}', \mathbf{b}) - \max_{\mathbf{b}} Q_j^j(\mathbf{x}', \mathbf{b}) \right) \right|}_{T_3}$$

Let us consider each term of the above equation individually to calculate the bound. For convenience, we can consider the following vector notations,

$$\mathbf{Q}_i^i = \left[ \max_{\mathbf{b}} Q_i^i(\mathbf{x}', \mathbf{b}), \ldots \right]^T \qquad \forall \mathbf{x}' \in \mathcal{X}$$

$$\mathbf{Q}_j^j = \left[ \max_{\mathbf{b}} Q_j^j(\mathbf{x}', \mathbf{b}), \ldots \right]^T \qquad \forall \mathbf{x}' \in \mathcal{X}$$

$$\mathbf{R}_i = \left[ r_i(\mathbf{x}, \mathbf{a}), \ldots \right]^T \qquad \forall \mathbf{x} \in \mathcal{X}, \mathbf{a} \in \mathcal{A}$$

$$\mathbf{R}_j = \left[ r_j(\mathbf{x}, \mathbf{a}), \ldots \right]^T \qquad \forall \mathbf{x} \in \mathcal{X}, \mathbf{a} \in \mathcal{A}$$

$$\mathbf{P}_i = \left[ p_i(\mathbf{x}'|\mathbf{x}, \mathbf{a}), \ldots \right]^T \qquad \forall \mathbf{x}' \in \mathcal{X}, \mathbf{a} \in \mathcal{A}$$

$$\mathbf{P}_j = \left[ p_j(\mathbf{x}'|\mathbf{x}, \mathbf{a}), \ldots \right]^T \qquad \forall \mathbf{x}' \in \mathcal{X}, \mathbf{x} \in \mathcal{A}$$

Using these notations, we can rewrite $T_2$ as the following,

$$\sum_{\mathbf{x}'} \left| (p_i - p_j) \max_{\mathbf{b}} Q_i^i(\mathbf{x}', \mathbf{b}) \right|$$

$$= \left\| (\mathbf{P}_i - \mathbf{P}_j) \cdot \mathbf{Q}_i^i \right\|_1$$

$$\leq \left\| \mathbf{P}_i - \mathbf{P}_j \right\|_1 \left\| \mathbf{Q}_i^i \right\|_\infty \qquad \text{using Hölder's inequality, } ||fg||_1 \leq ||f||_p ||g||_q \text{ where } \frac{1}{p} + \frac{1}{q} = 1$$

$$= 2\delta_{ij}^{TV} \left\| \mathbf{Q}_i^i \right\|_\infty \qquad \text{where } \delta_p^{ij} \text{ is the total variation distance, } \delta_{ij}^T = D_{TV}(\mathbf{P}_i, \mathbf{P}_j)$$

Similarly, we can write for $T_3$,

$$\sum_{\mathbf{x}'} \left| p_j \left( \max_{\mathbf{b}} Q_i^i(\mathbf{x}', \mathbf{b}) - \max_{\mathbf{b}} Q_j^j(\mathbf{x}', \mathbf{b}) \right) \right|$$

$$\leq \left\| \mathbf{P}_j \right\|_1 \left\| \mathbf{Q}_i^i - \mathbf{Q}_j^j \right\|_\infty$$

$$= \left\| \mathbf{Q}_i^i - \mathbf{Q}_j^j \right\|_\infty \qquad \text{because } \left\| \mathbf{P}_j \right\|_1 = 1$$

Thus we can write the following,

$$|Q_i^i(\mathbf{x}, \mathbf{a}) - Q_j^j(\mathbf{x}, \mathbf{a})| \leq \|\mathbf{R}_i - \mathbf{R}_j\|_\infty + 2\gamma \delta_{ij}^{TV} \mathbf{Q}_i^i + \gamma \left\|\mathbf{Q}_i^i - \mathbf{Q}_j^j\right\|_\infty$$

Because this is true for all $a \in \mathcal{S}, a \in \mathcal{A}$, we can write the following,

$$||\mathbf{Q}_i^i - \mathbf{Q}_j^j||_\infty \leq \|\mathbf{R}_i - \mathbf{R}_j\|_\infty + 2\gamma \delta_{ij}^{TV} \mathbf{Q}_i^i + \gamma \left\|\mathbf{Q}_i^i - \mathbf{Q}_j^j\right\|_\infty$$

$$\Rightarrow ||\mathbf{Q}_i^i - \mathbf{Q}_j^j||_\infty \leq \frac{\delta_{ij}^r}{1 - \gamma} + \frac{2\gamma \delta_{ij}^{TV}}{1 - \gamma} ||\mathbf{Q}_i^i||_\infty$$

$$\Rightarrow ||\mathbf{Q}_i^i - \mathbf{Q}_j^j||_\infty \leq \frac{\delta_{ij}^r}{1 - \gamma} + \frac{2\gamma R_{max,i} \delta_{ij}^{TV}}{(1 - \gamma)^2}$$

**(b)**

$$|Q_j^j(\mathbf{x}, \mathbf{a}) - Q_i^j(\mathbf{x}, \mathbf{a})|$$

$$= \left| r_j(\mathbf{x}, \mathbf{a}) + \gamma \sum_{\mathbf{x}'} p_j(\mathbf{x}'|\mathbf{x}, \mathbf{a}) \max_{\mathbf{b}} Q_j^j(\mathbf{x}', \pi_j(\mathbf{x})) - r_i(\mathbf{x}, \mathbf{a}) - \gamma \sum_{\mathbf{x}'} p_j(\mathbf{x}'|\mathbf{x}, \mathbf{a}) \max_{\mathbf{b}} Q_j^i(\mathbf{x}', \pi_j(\mathbf{x})) \right|$$

$$\leq |r_j(\mathbf{x}, \mathbf{a}) - r_i(\mathbf{x}, \mathbf{a})| + \gamma \left| \sum_{\mathbf{x}'} p_j(\mathbf{x}'|\mathbf{x}, \mathbf{a}) \max_{\mathbf{b}} Q_j^j(\mathbf{x}', \pi_j(\mathbf{x})) - \sum_{\mathbf{x}'} p_i(\mathbf{x}'|\mathbf{x}, \pi_i(\mathbf{x})) \max_{\mathbf{b}} Q_j^j(\mathbf{x}', \pi_j(\mathbf{x})) \right|$$

$$\leq |r_j(\mathbf{x}, \mathbf{a}) - r_i(\mathbf{x}, \mathbf{a})| + \gamma \sum_{\mathbf{x}'} \left| p_j \max_{\mathbf{b}} Q_j^j(\mathbf{x}', \pi_j(\mathbf{x})) - p_i \max_{\mathbf{b}} Q_j^i(\mathbf{x}', \pi_j(\mathbf{x})) \right|$$

$$\leq |r_j(\mathbf{x}, \mathbf{a}) - r_i(\mathbf{x}, \mathbf{a})| + \gamma \sum_{\mathbf{x}'} \left| p_j \max_{\mathbf{b}} Q_j^j(\mathbf{x}', \pi_j(\mathbf{x})) - p_i \max_{\mathbf{b}} Q_j^j(\mathbf{x}', \pi_j(\mathbf{x})) + p_i \max_{\mathbf{b}} Q_j^j(\mathbf{x}', \pi_j(\mathbf{x})) - p_i \max_{\mathbf{b}} Q_j^i(\mathbf{x}', \pi_j(\mathbf{x})) \right|$$

$$= |r_j(\mathbf{x}, \mathbf{a}) - r_i(\mathbf{x}, \mathbf{a})| + \gamma \sum_{\mathbf{x}'} \left| (p_j - p_i) \max_{\mathbf{b}} Q_j^j(\mathbf{x}', \pi_j(\mathbf{x})) + p_i \left( \max_{\mathbf{b}} Q_j^j(\mathbf{x}', \pi_j(\mathbf{x})) - \max_{\mathbf{b}} Q_j^i(\mathbf{x}', \pi_j(\mathbf{x})) \right) \right|$$

$$\leq |r_j(\mathbf{x}, \mathbf{a}) - r_i(\mathbf{x}, \mathbf{a})| + \gamma \sum_{\mathbf{x}'} \left| (p_j - p_i) \max_{\mathbf{b}} Q_j^j(\mathbf{x}', \pi_j(\mathbf{x})) \right| + \gamma \sum_{\mathbf{x}'} \left| p_i \left( \max_{\mathbf{b}} Q_j^j(\mathbf{x}', \pi_j(\mathbf{x})) - \max_{\mathbf{b}} Q_j^i(\mathbf{x}', \pi_j(\mathbf{x})) \right) \right|$$

$$\leq \|\mathbf{R}_i - \mathbf{R}_j\| + \gamma \|P_i - P_j\|_1 \left\|\mathbf{Q}_j^j\right\|_\infty + \gamma \|\mathbf{P}_i\|_1 \left\|\mathbf{Q}_j^j - \mathbf{Q}_i^j\right\|_\infty$$

$$\leq \delta_{ij}^r + \frac{2\gamma \delta_{ij}^{TV} R_{max,j}}{1 - \gamma} + \gamma \left\|\mathbf{Q}_j^j - \mathbf{Q}_i^j\right\|_\infty$$

Because this is true for all $\mathbf{x} \in \mathcal{X}, \mathbf{a} \in \mathcal{A}$, we can write the following,

$$||\mathbf{Q}_j^j - \mathbf{Q}_j^i||_\infty \leq \frac{\delta_{ij}^r}{1 - \gamma} + \frac{2\gamma \delta_{ij}^{TV} R_{max,j}}{(1 - \gamma)^2}$$

Finally, we can combine (a) and (b) to write the following,

$$||\mathbf{Q}_i^i - \mathbf{Q}_i^j||_\infty \leq \frac{2\delta_{ij}^r}{1 - \gamma} + \frac{2\gamma \delta_{ij}^{TV} (R_{max,i} + R_{max,j})}{(1 - \gamma)^2}$$

$\square$

| environment | change type | source $(\mathcal{S})$ | Target 1 $(\mathcal{T}_1)$ | Target 2 $(\mathcal{T}_2)$ | Target 3 $(\mathcal{T}_3)$ | Target 4 $(\mathcal{T}_4)$ |
|---|---|---|---|---|---|---|
| HalfCheetah-v3 | bthigh damping | 6.0 | 9.0 | 12.0 | 15.0 | 18.0 |
| | bshin damping | 4.5 | 6.0 | 9.0 | 12.0 | 15.0 |
| | bfoot damping | 3.0 | 3.0 | 6.0 | 9.0 | 12.0 |
| | fthigh damping | 4.5 | 9 | 12.0 | 15.0 | 18.0 |
| | fshin damping | 3.0 | 6.0 | 9.0 | 12.0 | 15.0 |
| | ffoot damping | 1.5 | 3.0 | 6.0 | 9.0 | 12.0 |
| Ant-v3 | fright upper length | 0.2 | 0.2 | 0.2 | 0.2 | 0.2 |
| | fright lower length | 0.4 | 0.5 | 0.2 | 0.3 | 0.3 |
| | fleft upper length | 0.2 | 0.2 | 0.2 | 0.2 | 0.2 |
| | fleft lower length | 0.4 | 0.5 | 0.2 | 0.3 | 0.3 |
| | bright upper length | 0.2 | 0.2 | 0.2 | 0.2 | 0.2 |
| | bright lower length | 0.4 | 0.5 | 0.2 | 0.5 | 0.3 |
| | bleft upper length | 0.2 | 0.2 | 0.2 | 0.2 | 0.2 |
| | bleft lower length | 0.4 | 0.5 | 0.2 | 0.5 | 0.5 |
| Humanoid-v3 | right shin length | 0.30 | 0.05 | 0.05 | 0.30 | 0.30 |
| | right upper arm | 0.277 | 0.277 | 0.139 | 0.277 | 0.277 |
| | left upper arm | 0.277 | 0.277 | 0.139 | 0.277 | 0.277 |
| | right shin size | 0.049 | 0.049 | 0.049 | 0.01 | 0.01 |
| | left shin size | 0.049 | 0.049 | 0.049 | 0.01 | 0.01 |
| | right foot size | 0.075 | 0.075 | 0.075 | 0.01 | 0.01 |
| | left foot size | 0.075 | 0.075 | 0.075 | 0.01 | 0.01 |
| | right lower arm size | 0.031 | 0.031 | 0.031 | 0.031 | 0.01 |
| | left lower arm size | 0.031 | 0.031 | 0.031 | 0.031 | 0.01 |
| | right hand size | 0.04 | 0.04 | 0.04 | 0.04 | 0.01 |
| | left hand size | 0.04 | 0.04 | 0.04 | 0.04 | 0.01 |

Table 2: Target task specifications

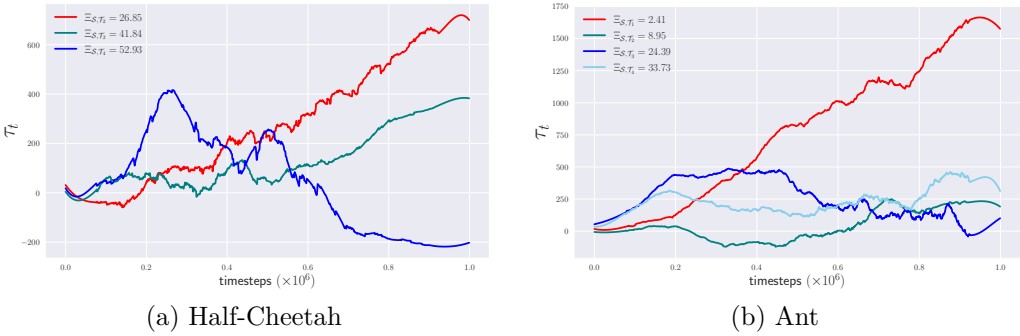

(a) Half-Cheetah                    (b) Ant

Figure 6: Relative transfer performance for REPAINT against PPO

## C   Experiment details

**HalfCheetah-v3:** This is a complex continuous control task of a 2D cat-like robot where the objective is to apply torque on the joints to make it run as fast as possible. The observation space is 17-dimensional and the action space, $a \in \mathbb{R}^6 \forall a \in \mathcal{A}$. Each action is a torque applied to one of the front or back rotors of the robot and can take a value between $[-1.0, 1.0]$. We make two types of perturbations to create target tasks; reward variation and dynamics variation. For the reward variation, the source environment uses a forward reward of $+1$, and the four target environments have a forward reward $r = [-2, -1, 1, 2]$. Note that a negative forward reward is a target task where the robot needs to run in the opposite direction than the source. For this type of example, the source acts as an adversarial task and the goal is to learn at least as good as learning from scratch. Next, we consider target tasks with varying dynamics. The source environment has the standard gym values for damping and the four target environments have different values of damping increased gradually in each task. The least similar task has the highest damping values in the joints.

**Ant-v3:** This is also a high dimensional continuous control task where the goal is to make an ant-robot move in the forward direction by applying torques on the hinges that connect each leg and torso of the robot. The observation space is 27-dimensional and the action space, $a \in \mathbb{R}^8 \forall a \in \mathcal{A}$ where each action is a torque applied at the hinge joints with a value between $[-1.0, 1.0]$. The source environment has the standard gym robot and the four target environments have varied dynamics by changing the leg lengths of the robot. Representative figures of these dynamics can be found in the appendix.

**Humanoid-v3:** This is a high-dimensional continuous control task where the goal is to make a humanoid robot move in the forward direction by applying torques on the hinge joints. The observation space is 376-dimensional and the action space, $a \in \mathbb{R}^{17} \forall a \in \mathcal{A}$ where each action is a torque applied at the hinge joints with a value between $[-0.4, 0.4]$. The source environment has the standard gym robot and the four target environments have varied dynamics by changing the hand and leg lengths as well as sizes.

## D   Algorithm hyperparameters

We keep the hyperparameters the same across all environments.

| parameter name | value |
|---|---|
| policy network hidden size | 200 |
| policy network layers | 4 |
| learning rate | 3e-4 |
| replay buffer size | 100000 |
| evaluation steps per epoch | 1000 |
| maximum episode length | 1000 |
| batch size | 64 |
| number of gradient updates | 50 |

## E   REPAINT algorithm

The authors utilizes the clipped loss from PPO algorithm(Schulman et al., 2017) for building the REPAINT algorithm,

$$L_{\text{clip}}(\theta) = \hat{\mathbb{E}}_t \left[ \min \left( l_\theta(\mathbf{x}_t, \mathbf{a}_t) \cdot \hat{A}_t, \text{clip}_\epsilon(l_\theta(\mathbf{x}_t, \mathbf{a}_t)) \cdot \hat{A}_t \right) \right] \quad \text{where} \quad l_\theta = \frac{\pi_\theta(\mathbf{a}_t|\mathbf{x}_t)}{\pi_{\text{old}}(\mathbf{a}_t|\mathbf{x}_t)}.$$

A modified version of this clipped loss is expressed using the ratio of target and source policy,

$$L_{\text{ins}}(\theta) = \hat{\mathbb{E}}_t \left[ \min \left( l_\theta(\mathbf{x}_t, \mathbf{a}_t) \cdot \hat{A}_t, \text{clip}_\epsilon(l_\theta(\mathbf{x}_t, \mathbf{a}_t)) \cdot \hat{A}_t \right) \right] \quad \text{where} \quad \rho_\theta = \frac{\pi_\theta(\mathbf{a}_t|\mathbf{x}_t)}{\pi_{\mathcal{S}}^*(\mathbf{a}_t|\mathbf{x}_t)}.$$

The details of the algorithm are provided in Algorithm 3.

---

**Algorithm 3** REPAINT algorithm (Tao et al., 2020)

---

1: **Initialize:** value network parameter $\nu$, policy network parameter $\theta$, source policy $\pi_{\mathcal{S}}^*$
2: **Set hyper-parameters:** $\zeta, \alpha_1, \alpha_2, \beta_k$
3: **for** $k = 1, 2, \ldots$ **do**
4:   Set $\theta_{\text{old}} \leftarrow \theta$
5:   Collect sample $\mathcal{S} = \{(\mathbf{x}, \mathbf{a}, \mathbf{x}', r)\}$
6:   Collect sample $\tilde{\mathcal{S}} = \{(\tilde{\mathbf{x}}, \tilde{\mathbf{a}}, \tilde{\mathbf{x}}', \tilde{r})\}$
7:   Fir state-value network, $V_\nu$ using only $\mathcal{S}$ to update $\nu$
8:   Compute advantage estimates $\hat{A}_1, \ldots, \hat{A}_T$ for $\mathcal{S}$ and $\hat{A}_1', \ldots, \hat{A}_T'$ for $\tilde{\mathcal{S}}$
9:   **for** $t = 1, \ldots, T'$ **do**
10:     **if** $\hat{A}_t < \zeta$ **then**
11:       Remove $\hat{A}_t'$ and the corresponding transition $(\tilde{\mathbf{x}}, \tilde{\mathbf{a}}, \tilde{\mathbf{x}}', \tilde{r})$ from $\tilde{S}$
12:     **end if**
13:   **end for**
14:   Compute sample gradient of $L_{\text{rep}}^k(\theta)$ using $\mathcal{S}$ where

$$L_{\text{rep}}^k(\theta) = L_{\text{clip}}(\theta) - \beta_k L_{\text{aux}}(\theta)$$

15:   Compute sample gradient of $L_{\text{ins}}(\theta)$ using $\tilde{\mathcal{S}}$ where

$$L_{\text{ins}}(\theta) = \hat{\mathbb{E}}_t[\min \rho_\theta(\mathbf{x}_t, \mathbf{a}_t) \cdot \hat{A}_t, \text{clip}_\epsilon(\rho_\theta(\mathbf{x}_t, \mathbf{a}_t)) \cdot \hat{A}_t']$$

16:   Update policy network by

$$\theta \leftarrow \theta + \alpha_1 \nabla_\theta L_{\text{rep}}^k(\theta) + \alpha_2 \nabla_\theta L_{\text{ins}}(\theta)$$

17: **end for**

---

## F   Soft Actor-Critic (SAC) algorithm

Soft actor-critic (SAC) is an off-policy model-free reinforcement learning algorithm. SAC builds upon the maximum entropy objective in RL where the optimal policy aims to maximize both the expected sum of returns and its entropy at each visited state. This can be expressed as the following

$$\pi^* = \arg\max_\pi \sum_t \mathbb{E}_{(\mathbf{x}_t, \mathbf{a}_t) \sim \rho_\pi}[r_t + \alpha \mathcal{H}(\pi(\cdot|\mathbf{x}_t))], \tag{9}$$

where $\rho_\pi$ is the state-action marginal of the trajectory distribution induced by the policy $\pi(\cdot|\mathbf{x}_t)$ and $\mathcal{H}(\cdot)$ is the entropy of the policy and $\alpha$ is a temperature parameter to control the effect of entropy. Using this objective it is possible to derive soft-Q values and soft-policy iteration algorithm as the following,

$$\mathcal{T}^\pi Q(\mathbf{x}_t, \mathbf{a}_t) = r_t + \gamma \mathbf{E}_{\mathbf{x}_{t+1} \sim p}[V(\mathbf{x}_{t+1})], \tag{10}$$

where $\mathcal{T}^\pi$ is the modified bellman backup operator and $V(\mathbf{x}_t) = \mathbf{E}_{\mathbf{a}_t \sim \pi}[Q(\mathbf{x}_t, \mathbf{a}_t) - \alpha \log \pi(\mathbf{a}_t | \mathbf{x}_t)]$.

Using several approximations to the soft-policy iteration algorithm it is possible to obtain an actor-critic architecture that maximizes the objective in Eq. 9. This is specifically done by using deep neural networks to parameterize the value function and the policy. Finally, the objective function can be re-written as the following,

$$J(\phi) = \mathbb{E}_{\mathbf{x}_t \sim \mathcal{D}} \left[ D_{\mathrm{KL}} \left( \pi_\phi(\cdot | \mathbf{x}_t) \,\middle\|\, \frac{\exp\left(Q_\theta(\mathbf{x}_t, \cdot)\right)}{Z_\theta(\mathbf{x}_t)} \right) \right] \tag{11}$$

| Symbol | meaning |
|---|---|
| $\mathcal{M}, \mathcal{M}_\mathcal{S}, \mathcal{M}_\mathcal{T}$ | MDP, source MDP, target MDP |
| $\mathcal{S}$ | source task |
| $\mathcal{T}$ | target task |
| $\mathcal{X}, \mathcal{A}$ | state space, action space |
| $\mathcal{R}_\mathcal{S}, \mathcal{P}_\mathcal{S}$ | source reward, source transition dynamics |
| $\mathcal{R}_\mathcal{T}, \mathcal{P}_\mathcal{T}$ | target reward, target transition dynamics |
| $\mathcal{K}_\mathcal{S}$ | source knowledge |
| $\mathcal{K}_\mathcal{T}$ | target knowledge |
| $\mathcal{D}_\mathcal{T}$ | dataset collected in target task |
| $\mathcal{H}(\cdot, \cdot)$ | cross-entropy |
| $\pi_\mathcal{S}^*$ | optimal source policy |
| $\pi_\mathcal{T}$ | target policy |
| $A_\mathcal{S}, A_\mathcal{T}$ | advantage using source policy, advantage using target policy |
| $\rho$ | transfer evaluation metric |
| $\tau$ | relative transfer performance |
| $\Lambda_i$ | transferability of algorithm $i$ |
| $G_t$ | returns at timestep $t$ |
| $\Xi_{\mathcal{S}, \mathcal{T}}^\mathcal{P}$ | dyanmics similarity between $\mathcal{S}, \mathcal{T}$ |
| $\Xi_{\mathcal{S}, \mathcal{T}}^\mathcal{R}$ | reward similarity between $\mathcal{S}, \mathcal{T}$ |
| $\beta$ | temperature parameter |
| $\psi$ | source policy parameters e.g. $\pi_\mathcal{S}^* \equiv \pi_\psi$ |
| $\phi$ | target policy parameters e.g. $\pi_\theta$ |
| $\theta$ | value function parameters |

Table 3: Nomenclature

