# OpenReview forum: "An advantage based policy transfer algorithm for reinforcement learning with measures of transferability"
_TMLR — Rejected by TMLR_

### Review · Reviewer_NMr9 · 2025-03-17

**Summary Of Contributions:**

This paper studies the cross-domain transfer learning in RL (under the assumption that both the source and target domains have the same state space and action space) from the perspective of domain transferability. Specifically, the authors propose to use the advantage evaluated on actions selected by the source policy as a regularization coefficient that reflects the influence of the source on the target. This idea leads to a new algorithm APT-RL, which achieves policy transfer via policy regularization with an adaptive temperature parameter. Moreover, this paper also describes the notion of transferability, including a measurement method for model-based task similarity.
The proposed APT-RL algorithm is evaluated on multiple MuJoCo locomotion tasks against several baselines, including the zero-shot source policy, the source policy fine-tuned with target data, and REPAINT.

**Audience:**

Yes

**Broader Impact Concerns:**

Currently there is no Broader Impact statement in this paper.

**Claims And Evidence:**

Yes

**Requested Changes:**

Here are the questions to be addressed to strengthen this work:
1. What are the new insights provided in the discussion about transferability in Section 4.1? It appears that Definition 4.1 essentially states that one can define a mapping from source-domain and target-domain knowledge to some real value, but it does not provide anything specific.
2. Can APT-RL handle the transfer scenarios where the structure of the domains are similar but the corresponding optimal policies are fairly different (in terms of cross entropy)? Please also refer to the Gridworld example in the above.
3. How does APT-RL perform compared to the more recent benchmark methods, such as DARC [1], VGDF [2], IGDF [3], and PAR [4]? Moreover, it would be good to discuss these works in the Related Work.

**Strengths And Weaknesses:**

**Strengths**
- The viewpoint of rethinking policy transfer as policy regularization appears interesting. Moreover, the advantage-based approach of choosing the proximal / regularization term is conceptually simple and reasonable.
- Overall the paper is clearly-written, with sufficient motivation and justification in most places.
- APT-RL empirically exhibits substantial performance improvement over several standard RL transfer methods on locomotion tasks.

**Weaknesses**
- One major concern is that the policy regularization approach can only address a subcollection of the RL transfer scenarios. Take the Gridworld example in the VGDF paper (Xu et al., 2023) https://proceedings.neurips.cc/paper_files/paper/2023/hash/e8ad87f1076fb0f75d89a45828f186b0-Abstract-Conference.html. While the source and target domains have similar layouts and hence shall enjoy good transfer, their optimal policies can differ by a lot in terms of cross entropy. Therefore, using policy regularization can be somewhat limited as it cannot address such cases. Indeed, the target domains considered in the experiments (Section 6) are mostly obtained by adding some variations to the parameters of source domains.
- Regarding the measure of transferability in Section 4.1, it is somewhat unclear what the new insights are in this part. Definition 4.1 essentially states that one can define a mapping from source-domain and target-domain knowledge to some real value, but it does not provide anything specific. To make Definition 4.1 more meaningful, it shall at least concretely lead to a few such mappings that can connect the transferability measure to the actual domain gap. Moreover, the technical support of Theorem 4.1 also appears straightforward and does not add much to it. Please correct me if I missed anything.
- Theorem 4.2 appears to be a direct application of the widely used Simulation lemma, e.g., see Lemma 1 and Lemma 2 in this note (https://nanjiang.cs.illinois.edu/files/cs598/note3.pdf). However, there is nearly zero discussion about the connection between these two results.
- The baselines considered in this paper are not up to date. There are several more recent cross-domain RL methods missing in the comparison, such as DARC [1], VGDF [2], IGDF [3], and PAR [4]. All the above focus on transferring a source-domain policy to another target domain with different environment dynamics and reward functions. To showcase the benefit of using APT-RL, a comparison with these more recent methods in both empirical evaluation and algorithm design is needed.

[1] Eysenbach et al., “Off-dynamics reinforcement learning: Training for transfer with domain classifiers,” ICLR 2020.

[2] Xu et al., “Cross-Domain Policy Adaptation via Value-Guided Data Filtering,” NeurIPS 2023.

[3] Wen et al., “Contrastive Representation for Data Filtering in Cross-Domain Offline Reinforcement Learning,” ICML 2024.

[4] Lyu et al., “Cross-Domain Policy Adaptation by Capturing Representation Mismatch,” ICML 2024.

---

> ### Author Response · Authors · 2025-05-06
> **Response to reviewer NMr9**
>
> We would like to thank reviewer NMr9 for providing valuable comments and showing connection to important literature that we missed earlier. In the following we address each comment of reviewer NMr9:
>
> *Comment 1: What are the new insights provided in the discussion about transferability in Section 4.1? It appears that Definition 4.1 essentially states that one can define a mapping from source-domain and target-domain knowledge to some real value, but it does not provide anything specific.*
>
> **response:**   We thank the reviewer for this point. We intentionally keep the definition of transferability measure open-ended so that we can unify different transfer measures from literature. Definition 4.1 is a generalized version of the existing measures. For example, in Table 1, we demonstrate concrete examples of various mappings. Specifically, we show that our concept of transferability measure unifies different types of knowledge transfer used in literature such as average returns, samples required to obtain asymptotic returns, area under the reward curve etc.
>
> For the technical support of Theorem 4.1, we would like to argue that this Theorem provides important context in evaluating transfer strategy. Using this simple idea, one can identify whether a policy in the target task will be at least as good as learning from scratch by just looking at the relative transfer measure. Additionally, this relative transfer measure helps us to compare between multiple different transfer strategies without manually inspecting the learning from scratch. We demonstrate this idea in Figure 4. For example, the relative transfer measure for HalfCheetah in Figure 4 shows that transferring source knowledge is beneficial to all three target tasks.
>
> *Comment 2: Can APT-RL handle the transfer scenarios where the structure of the domains are similar but the corresponding optimal policies are fairly different (in terms of cross entropy)? Please also refer to the Gridworld example in the above.*
>
> **response:**  We thank the reviewer for pointing out this important information and the related literature. We agree that the task is challenging when the optimal policy differs greatly between the source and the target. We would like to mention that our work proposes two strategies to overcome this:
>
> 1. One of the key benefits of APT-RL is that it performs similarly to learning from scratch against an adversarial source task. When the target optimal policy differs a lot from the source optimal policy then APT-RL puts less weight to source knowledge. This can be seen in Figure 3 (d) for half-cheetah and Humanoid.
> 2. We also show that for practical application, task similarity might provide information on whether a transfer is feasible or not. The purpose of figure 2 is to demonstrate that less similar tasks might have optimal policies that differ a lot.
>
> To incorporate the suggestions of the reviewer, we have also updated section 7 (limitations) with related discussions.
>
> *Comment 3: How does APT-RL perform compared to the more recent benchmark methods, such as DARC [1], VGDF [2], IGDF [3], and PAR [4]? Moreover, it would be good to discuss these works in the Related Work.*
>
> **response:**  We thank the reviewer for the related literature. We would like to mention that DARC, VGDF, IGDF and PAR are cross-domain policy adaptation techniques and all of them collects data from the source task during policy update. This is in contrast to our approach where we directly transfer the policy from the source task and do not allow source data collection. Our algorithm only collects data from the target task and thus comparison with these methods is out of the scope of this paper. We leave this as an important future direction where policy transfer can be compared with sample transfer in a more rigorous setting.
>
> We have taken the reviewer's suggestion and added a new paragraph in section 2 (related work), where we discussed the recently proposed DARC, VGDF, IGDF and PAR algorithms and how our problem formulation differs from these studies.
>
> *Comment 4:  Theorem 4.2 appears to be a direct application of the widely used Simulation lemma, e.g., see Lemma 1 and Lemma 2 in this note (https://nanjiang.cs.illinois.edu/files/cs598/note3.pdf). However, there is nearly zero discussion about the connection between these two results.*
>
> **response:**  This is an important point and based on the reviewer's suggestion we have referenced the simulation lemma in section 4.2 and discussed its relation with our results. More specifically, the simulation lemma shows the value-function bound between two MDPs for any policy where one MDP is a sufficient approximation of another MDP. Relatedly, we demonstrate the action-value bound between target MDP with target optimal policy and target MDP with source optimal policy. Our results is certainly motivated by this literature and we thank the reviewer for addressing this.

---

> ### Comment · Reviewer_NMr9 · 2025-05-25
>
> Thank the authors for the detailed response and the revised paper. Some of my questions have been addressed, such as the recent cross-domain RL baselines and the simulation lemma. I still have the following concerns and a few follow-up questions:
>
> 1. **Regarding the discussion about transferability in Section 4.1**: I appreciate the added examples, which can help motivate the definition of transferability. That being said, as the paper only designs an algorithm for one specific case (i.e., the source optimal policy is the the knowledge to be transferred, data collected from the target task is used as the target knowledge, and average returns during evaluation are used as the performance metric), I still find it somewhat unnecessary to have the discussion about transferability in Section 4.1.
>
> 2. **Regarding the transfer scenarios where the structure of the domains are similar but the corresponding optimal policies are fairly different**: Thank the authors for confirming this limitation of APT-RL, and it is good to see that the authors make this limitation explicit in the revised version, which can help the readers better understand the benefits and limitations of APT-RL.
> However, this indeed makes APT-RL less applicable in practice since the discrepancy between the source-domain and target-domain optimal policies is typically unknown a priori (and such a scenario is very common, e.g., navigation in an environment with a quite different layout, robot locomotion with different morphologies, robot arm manipulation with a different embodiment). This issue is also mentioned by Reviewer 3BJ1.
>
>     Moreover, such scenarios can be addressed by REPAINT (Tao et al., 2021), as shown in the experiments on both MuJoCo and AWS DeepRacer in Section 6 of (Tao et al., 2021).. Based on the above, it would be good to evaluate APT-RL on these tasks (say AWS DeepRacer) to better check the performance of APT-RL when source and target optimal policies substantially differ.

---

> > ### Author Response · Authors · 2025-06-04
> > **Additional response to reviewer NMr9**
> >
> > We sincerely thank the reviewer for carefully reading our rebuttal and the revised paper. We appreciate the time and thoughtful feedback, which have contributed meaningfully to improving the quality of our work. In the following, we address each of the additional concerns of mentioned by the reviewer.
> >
> >
> >  *Comment 1: Regarding the discussion about transferability in Section 4.1*
> >
> > **response:** We appreciate the reviewer’s feedback. We included Section 4.1 to provide a broader conceptual framework that helps unify and contextualize various transfer learning approaches. While our experiments focus on a specific case, transferring a source optimal policy, using target task data, and measuring performance via average returns, our formulation of transferability in Section 4.1 is general and applicable across different settings. We choose this specific case to evaluate the transferability rigorously across several tasks.
> >
> > As shown in Table 1, this framework allows us to map and compare diverse transfer methods in a consistent way. We believe this adds clarity to the landscape of transfer learning and strengthens the theoretical foundation of our approach, even if the experiments focus on one instance of the broader concept.
> >
> >
> > *Comment 2: Regarding the transfer scenarios where the structure of the domains are similar but the corresponding optimal policies are fairly different:*
> >
> > **response:** We would like to argue that the problem setting does not make Apt-RL less applicable in practical scenario. In fact, our problem setting provides additional flexibility in algorithm design, as we do not make explicit assumption that we have access to a source or source simulator. Our problem setting is also supported extensively by previous literature. For example, transfer of some sort of features (e.g. successor features) have been applied [1, 2]. Previous literature also made assumptions that state distribution of an optimal policy in the source domain will resemble the state distribution of an optimal policy in the target domain [3]. In practical scenario, often only the source policy is available that can be transferred to a target task and thus Apt-RL is mostly suitable for sim2real or behavior transfer scenarios where the target task may change in terms of dynamics and/or rewards.
> >
> > We agree with the reviewer that the discrepancy between source and target is unknown apriori. This is precisely why we proposed task similarity approaches discussed in section 4.1 to quickly identify the nature of the target task using small number of samples. From a practical perspective, this task similarity approach can provide insights on whether the transfer of policy might be useful for a certain target task or not. We would like to clarify that APT-RL has already been evaluated against the REPAINT algorithm in all the tasks reported in our paper. Our focus is on high-dimensional continuous control tasks, which are generally more challenging than the discrete or lower-dimensional tasks used in REPAINT’s evaluation (e.g., reward variation in simple MuJoCo setups). In contrast, our benchmarks involve twelve tasks across three MuJoCo environments with significant dynamics variations.
> >
> > As shown in Figure 3, APT-RL consistently outperforms REPAINT by a large margin and is able to solve complex tasks such as those in the Humanoid environment, where REPAINT fails. Therefore, we argue that our current evaluation setting sufficiently demonstrates the superiority of APT-RL over REPAINT, especially in the context of high-dimensional control, which is the central focus of this paper.
> >
> > [1] André Barreto, Will Dabney, Rémi Munos, Jonathan J Hunt, Tom Schaul, Hado P van Hasselt, and David
> > Silver. Successor features for transfer in reinforcement learning. Advances in neural information processing
> > systems, 30, 2017
> >
> > [2] Campos, V., Sprechmann, P., Hansen, S., Barreto, A., Kapturowski, S., Vitvitskyi, A., ... \& Blundell, C. (2021). Beyond fine-tuning: Transferring behavior in reinforcement learning. arXiv preprint arXiv:2102.13515.
> >
> > [3] Abhishek Gupta, Coline Devin, YuXuan Liu, Pieter Abbeel, and Sergey Levine. Learning invariant feature
> > spaces to transfer skills with reinforcement learning. arXiv preprint arXiv:1703.02949, 2017.

---

### Review · Reviewer_h9eJ · 2025-04-10

**Summary Of Contributions:**

This paper proposes APT-RL, designed to improve the transfer of knowledge from a source environment to a target environment in fixed-domain settings.

The key idea is to use the advantage function as a regularization weight to control how much knowledge from the source policy should influence the target policy. The paper introduces a new transferability evaluation metric and a model-based task similarity measurement algorithm to assess how suitable a source task is for transfer.

Experiments demonstrate that APT-RL outperforms existing transfer RL algorithms and achieves comparable performance to learning from scratch in adversarial settings.

**Audience:**

Yes

**Claims And Evidence:**

No

**Requested Changes:**

1. In the definitions of the returns $G_t$ and $\rho_t$, there is no discount factor $\lambda$ and no termination signal, which could result in an infinite return. Could the author clarify the rationale behind this design choice? Furthermore, in Theorem 2, the bound includes a discount factor, and in the appendix proof, the author explicitly considers a discount factor in the target. This theoretical inconsistency must be addressed.

2. The author should briefly explain in the main paper how they created tasks with varying dynamics and rewards.

**Strengths And Weaknesses:**

**Strength**

1. The authors propose interesting relative transfer performance measure and scalable task similarity measurement algorithm, providing useful tools for analysing transfer RL performance across tasks.

2. The proposed APT-RL algorithm demonstrates strong performance compared to several baselines across a diverse set of high-dimensional continuous control tasks.

**Weaknesses**

I have not identified any clear weaknesses (math is not thoroughly examined).

---

> ### Author Response · Authors · 2025-05-06
> **Response to reviewer h9eJ**
>
> We thank the reviewer for providing insightful comments and catching important points. These comments have helped us improve the technical clarity of the paper. We address these comments in the following:
>
> *Comment 1:* In the definitions of the returns $G_t$ and $\rho_t$, there is no discount factor $\lambda$, and no termination signal, which could result in an infinite return. Could the author clarify the rationale behind this design choice? Furthermore, in Theorem 2, the bound includes a discount factor, and in the appendix proof, the author explicitly considers a discount factor in the target. This theoretical inconsistency must be addressed.
>
> **response:**  We thank the reviewer for this catching this important issue. We agree that the returns $G_t$ and $\rho_t$ should include the discount factor. We have taken the reviewer's suggestion into consideration and made corrections to section 4.1 with this discussion.
>
> *Comment 2:* The author should briefly explain in the main paper how they created tasks with varying dynamics and rewards.
>
> **response:** Based on the reviewer's comment we have added brief discussion about the tasks in section 5. We have also directed the reader to the appendix which contains details about how each task is created with associated numeric value for each parameter.

---

### Review · Reviewer_3BJ1 · 2025-04-11

**Summary Of Contributions:**

The paper proposes a policy transfer algorithm for fixed domain environments that weighs two objectives: the policy update on the target task (here SAC objective) and the cross-entropy loss between the source and target policy with a parameter derived from the difference in the advantage for target and source. The weight is not a hyperparameter but is set depending on the advantage functions. The proposed approach is evaluated on three Mujoco tasks where it performs comparably or slightly better than standard SAC.

The paper is not easy to follow especially due to re-occuring notation inaccuracies and a rather confusing introduction.

**Audience:**

Yes

**Claims And Evidence:**

No

**Requested Changes:**

- The minimal example seems not too related to the experiments, and it is unclear to me why one would even consider training these targets with different policies (instead of viewing S, T_1, and T_2 as 3 configurations of the environment). I believe that changing the "doors" instead of the goal location would make the example more realistic and closer to the presented experiments.
- The plots in Fig. 1 are difficult to relate to the environment. Please highlight the relationship between the plots and the gridworlds better.
- The second paragraph in the introduction is very long and unclear. What is a formal way to quantify? What is formal about the quantification? Can you separate the description/intuition from Fig 1 better from your contribution?
- The third paragraph in the introduction is also not too clear. Maybe start with the example or at least revise the first sentence to be crisp and clear. Generally, it is not too clear on page two for which task the advantage is calculated so I would already introduce a subscript or superscript as you do later in Sec. 3
- The contribution list is not to the point. How does the work extend existing work? Why do you comment on the motivation for task similarity measurement in the contribution? What is the key takeaway from the experiments?

- Please check your notation thoroughly. Here is what I think is not correct:
* \alpha in Eq 2 is not introduced
* \alpha_pi should be given in Alg. 1
* The equation in def. should be having domains only - \mathcal{K} seem to be domains but \rho is a scalar so this should be \mathbb{R}, also not correct in Sec. 5 \Gamma_i
* Why is \mathcal{M}_S and \mathcal{M}_T introduced
* Alg. 2 is all over the place - why do you learn the dynamics of the target and not the source? The super- and subscripts of f are not consistent
* Appendix, Theorem 1 has many obvious typos, e.g., brackets missing, and not introduced subscripts.

- Please clarify what is assumed to be known or APT-RL. I found it confusing that for the task similarity everything (including the reward function) is learned.
- The relationship between the third column and the 1. and 2. in Tab. 1 is not clear
- For sec. 4.2.1, it is not clear what is adapted from existing work and why the bounds motivate the algorithm. Please clarify the connection to the algorithm or remove it.
- I do not see a clear improvement over SAC in sample efficiency. I would have expected that. Why is that not the case?
- Looks like only dynamics transferability is investigated but not the reward transferability for the second set of half-cheetah tasks. Why is that?
- Fig. 4 needs more explanation. Why are similarities 0 at the beginning of the training? Why is similarity oscillating? The zoom-in for \beta is difficult to read; consider moving it next to the plot.

Minor:
- Since the supplementary material is just a pdf, I would prefer the content to be in the main document after the references

**Strengths And Weaknesses:**

Strengths:
- The derivation of $\beta\_t$ is innovative and seems to be close to optimal empirically
- Advantage-based decision of similarity is intuitive and application independent

Weaknesses:
- The paper does not clearly scope the problem addresses, i.e., motivation is not to the point, assumptions on reward function and simulation environment availability are unclear
- The notation is not consistently used and there is a significant amount of typos that reduces the readability of boh central algorithms significantly.
- The similarity measure introduction rather disrupts the paper than supports since it seems to  mainly an evaluation metric but is not as clearly framed as one.

---

> ### Author Response · Authors · 2025-05-06
> **response to reviewer 3BJ1 [part 1/2]**
>
> We would like to thank the reviewer for reading our work and providing detailed comments. We have found the comments very useful in improving the quality of the paper. In the following, we address each comment by the reviewer.
>
> 1. We would like to clarify that the goal of the toy example is to demonstrate the motivation behind the main ideas proposed in this paper such as: advantage based policy transfer, transfer evaluation measure, and importance of task similarity measurement. Note that one of the door positions is changed in both target tasks $\mathcal{T}_1$ and $\mathcal{T}_2$ compared to the source task $S$. Figure 1(d) shows that knowledge transfer is more effective in target $\mathcal{T}_1$ when compared to target $\mathcal{T}_2$. Figure 1(e)-1(f) show the intuition behind the idea of advantage based regularization coefficient for policy transfer. These plots show that Target $\mathcal{T}_1$ is more influenced by the source when compared to $\mathcal{T}_2$. Based on this comment, we have modified the introduction to better reflect this intuition.
>
> 2. We thank the reviewer for mentioning this issue. Based on this feedback, we have reorganized figure 1 that reflects the relationship between the gridworld and the plots.
>
> 3. We have made significant changes to the second paragraph of introduction to make it more concise. Note that we describe the idea in a formal way in section 3 and provide evaluation criteria in section 4. We have also modified the introduction further to better reflect the intuition and contributions.
>
> 4. We have made significant changes to the third paragraph for better readability. In addition, we added subscript to identify the task associated with the notation.
>
> 5. We have modified the contributions based on the reviewer's suggestion.
>
> 6. We have the edited the text below equation 2 to explain $\alpha$ which is the entropy regularization coefficient.
>
> 7. We have included this information in Alg. 1. Additionally, to avoid confusion between learning rate and entropy regularization coefficient, we have changed all notations of learning rate. For example, target learning rate is now $\delta_\pi$ instead of $\alpha_\pi$, source learning rate is now $\delta_\mathcal{S}$ instead of from $\alpha_\mathcal{S}$.
>
> 8. This is an important point and we have fixed this in the definition. We have also omitted $\Gamma_i$ in section 5.
>
> 9. $\mathcal{M}_S$ and $\mathcal{M}_T$ are introduced so that the definition of transferability is consistent with the rest of the paper in terms of the source task and target task respectively.
>
> 10. We thank the reviewer for catching the notation issues. Based on this comment, we have modified Alg. 2 and ensured that the revised notations are correct. We would like to clarify that we learn both the source and the target dynamics in Alg. 2.
>
> 11. We have edited the theorem and fixed the typos.
>
> 12. We would like to clarify that, only the source policy is assumed to be known for APT-RL. For task similarity, we assume that we can only interact with the environment and neither the reward function nor the dynamics model is known *apriori*. The motivation behind this assumption is to develop a simple method to quantify the similarity between two tasks by only interacting in those environments. For interaction, we choose a random policy to collect data which is supported by literature [1]. Finally, our ultimate motivation is to show that the transferability measure intuitively matches the similarity between the source and target tasks as shown in Figure 4.
>
> 13. The purpose of table 2 is to show that our proposed concept of transferability measure can unify previous transfer RL methods from literature. Based on the reviewer's comment, we have re-organized Table 2 by separating column 3 into a new table for clarity.
>
> 14.  We have updated section 4.2.1 to clarify the connection between existing literature and our work. Additionally, we would like to argue that section 4.2.1 provides important context for knowledge transfer in fixed domain environments. More specifically, this section explains the discrepancy between action value functions in the target task when we execute the target optimal policy and source optimal policy respectively. Intuitively, this bound helps us to understand the effect of executing source optimal policy directly in the target without any modifications. The action value bound depends on the reward and dynamics variations between the source and the target which act as the main motivation to develop the task similarity measurement algorithm.
>
>
> [1] Zhang, Amy, Harsh Satija, and Joelle Pineau. "Decoupling dynamics and reward for transfer learning." arXiv preprint arXiv:1804.10689 (2018).

---

> > ### Author Response · Authors · 2025-05-06
> > **response to reviewer 3BJ1 [part 2/2]**
> >
> > 15.  We would like to argue that APT-RL demonstrates clear improvements in sample efficiency compared to SAC, particularly in target tasks that are more closely aligned with the source task. As shown in Fig. 3 (half-cheetah), APT-RL outperforms SAC in $\mathcal{T}_1$ and $\mathcal{T}_2$, where the target dynamics share greater similarity with the source. In contrast, the improvement is marginal in $\mathcal{T}_3$, which diverges more significantly from the source, resulting in performance closer to learning from scratch. A similar trend is observed in the adversarial task $\mathcal{T}_4$, where transfer becomes less beneficial.  This pattern consistently appears across other agents in Fig. 3, including ant and humanoid. For instance, APT-RL shows improved sample efficiency in $\mathcal{T}_1$ through $\mathcal{T}_3$, while $\mathcal{T}_4$, being more adversarial or dissimilar, again shows comparable performance to SAC. Overall, these results suggest that APT-RL is most effective when the target task retains some structural or dynamic similarity to the source. As task similarity decreases, the advantage of transfer diminishes, leading to performance that converges with direct learning.
> >
> > 16. Thank you for mentioning this important point. We restricted reward‑transfer experiments to the half‑cheetah suite because, in standard Gym tasks, altering the reward function often turns the target into an adversarial variant, causing all algorithms—APT‑RL and baselines alike—to behave as if training from scratch. This gives little additional insight and leads to repetitive results across environments, whereas varying the dynamics offers a richer, more discriminative test of transferability. To keep the evaluation focused and non‑redundant, we therefore emphasized dynamics variation in the other settings while still including a reward‑shift example in half‑cheetah for completeness.
> >
> > 17. We would like to clarify that Fig. 4 shows relative transfer performance for each environment. Each color represents the corresponding mean task similarity from Fig. 2. As a result, the relative transfer performance starts from zero and not the similarity. As we do not transfer any knowledge initially, it starts from zero. Additionally, the relative transfer performance is oscillating in Fig. 4 and not the similarity. The oscillation can be explained by how APT-RL is setup. As we improve the source optimal policy with target data, the source policy becomes better over time and often provide good reward for some actions while bad rewards for other actions. As a result, APT-RL autonomously switches between higher and lower values of the regularization co-efficient for source policy. This phenomenon causes the oscillation in the transfer performance. We would like to note that, although the transfer performance oscillates, the net transfer performance curve is above the x-axis which means that knowledge transfer is effective. Additionally, we anticipate that, for humanoid environment we observe higher oscillation due to the complexity of the task.

---

> ### Comment · Reviewer_3BJ1 · 2025-05-14
> **Reply to Authors**
>
> Thanks for your response and the revised paper. In particular, the contributions are significantly clearer now. I have a few follow-up questions and comments:
>
> - I don't find the door difference between S and T_1 and T_2 convincing in the motivating example since the source tasks should have the same optimal policy with and without the door. Or is the initial state and goal not fixed for a task? I would encourage you to improve this example, e.g. using T_1 as source and moving the second or first door
> - Unfortunately, I do not perceive a clear benefit in sample efficiency for APT-RL from Fig. 2. For the Ant environment, I agree that APT-RL is slightly more sample-efficient with decreasing effect with more dissimilar tasks. For the HalfCheetah and Humanoid, the convergence of SAC and APT-RL does not seem to differ, i.e., the same sample efficiency. For T_1 and T_2 of the HalfCheetah, I see a significant difference in the final reward, while for all other experiments, the confidence intervals largely overlap. Please clarify why you interpret this differently or clarify your used terminology.
>
> Minor comments:
> - Please check the revised introduction for typos: fund mental, 1) 2), Figure 1(3)
> - I still would appreciate the supplementary material to be added to the main pdf as an appendix. Especially, because the proof of Theorem 2 is so briefly summarized (no intuition given) that one has to open the additional pdf.

---

> > ### Author Response · Authors · 2025-06-05
> > **Response to follow-up comments from reviewer 3BJ1**
> >
> > We thank the reviewer for reading our response and providing important insights. We address each comment in the following:
> >
> > **Response 1:** We thank the reviewer for this thoughtful question. We would like to clarify that the purpose of the toy example in Figure 1 is not to present a rigorous transfer learning scenario, but rather to provide a simple and intuitive illustration of our core idea, how the advantage-based regularization coefficient modulates the influence of the source policy. For rigorous evaluation of our algorithm we create extensive list of transfer learning tasks in Figure 3.
> >
> > To address the reviewer’s point: while the tasks may appear to share the same optimal policy structure, the critical distinction lies in the goal location, which is different in each target task ($T_1$ and $T_2$). These differences, though minor, are sufficient to demonstrate how the regularization adapts to varying task similarity. Importantly, the initial state and goal are fixed within each task but vary across tasks.
> >
> > Specifically, in $T_2$, the goal is moved farther away compared to $T_1$, making the source policy less useful in the target environment or only useful for a shorter portion of the trajectory. This naturally results in a lower regularization coefficient, as observed in Figure 1(f), reflecting the reduced transferability of the source policy. In contrast, $T_1$ remains more aligned with the source task, resulting in a higher influence of the source policy, as shown in Figure 1(e).
> >
> > We also considered the reviewer’s suggestion to modify the setup (e.g., changing the source to $T_1$ or repositioning a door), but we believe doing so would add unnecessary complexity and detract from the core intuition. The current setup provides a minimal and controlled scenario that clearly illustrates how our method adjusts the influence of the source policy based on task similarity. Based on this early motivation, we describe the technical details behind this intuition throughout the rest of the paper. We hope this clarifies the intent behind the example, and we appreciate the opportunity to elaborate on this further.
> >
> > **Response 2:** We explain our results for each task for clarity.
> >
> > Ant environment: In tasks $T_1$ and $T_2$, while APT-RL and SAC eventually reach similar final rewards, APT-RL accumulates high rewards significantly faster than SAC. This jumpstart happens due to the initial larger influence of the source policy which demonstrates improved sample efficiency. In $T_3$, APT-RL accumulates high reward much earlier than SAC and maintains the reward throughout the training period unlike SAC. This shows the sample efficiency of APT-RL in achieving similar of higher rewards than SAC. Finally, in $T_4$, APT-RL achieves slightly higher rewards than SAC. Note that this is due to the complexity of the task and as similarity decreases with the source, $T_4$ acts as an adversarial tasks and APT-RL still performs on-par with SAC.
> >
> > Humanoid environment: In $T_1$ and $T_2$, APT-RL shows both higher final reward and faster reward accumulation compared to SAC. In $T_3$, the benefit is even clearer; APT-RL rapidly achieves high rewards and maintains them consistently, while SAC progresses slowly. In $T_4$, the final performance of both methods appears similar, but APT-RL achieves slightly higher reward than SAC. This task is designed to be more complex (adversarial), making transfer harder; yet APT-RL still matches or exceeds SAC. This robustness is also reflected in the performance of the fine-tuned policy, which requires substantially more time to learn.
> >
> > Half-cheetah: In $T_1$ and $T_2$, we agree with the reviewer that APT-RL achieves better final rewards more quickly than SAC, clearly reflecting better sample efficiency. For $T_3$, both methods converge to similar final rewards, but APT-RL does so significantly faster by reaching SAC-level rewards around $2 \times 10^5$ timesteps, whereas SAC takes around $4 \times 10^5$. $T_4$ is explicitly designed to be an adversarial case (by reversing the direction of movement), where source policies are less helpful. Here, APT-RL performs comparably to SAC, showing that it degrades gracefully and does not underperform even in challenging transfer scenarios.
> >
> > In summary, we interpret sample efficiency not only in terms of the final reward achieved, but also based on how quickly that reward is reached and how consistently it is maintained throughout training. This interpretation aligns with established transfer evaluation in the reinforcement learning literature [1].
> >
> > [1] Matthew E Taylor and Peter Stone. Transfer learning for reinforcement learning domains: A survey. Journal
> > of Machine Learning Research, 10(7), 2009.
> >
> > **Response to minor comments:** We particularly thank the reviewer for addressing these issues.
> > 1. We have fixed the typos in the updated draft
> > 2. We agree with the reviewer and included the supplementary materials with the main paper

---

### Decision · Action_Editor_SPnu · 2025-06-08

**Recommendation:** Reject

**Audience:**

Yes

**Audience Explanation:**

The paper introduces a novel approach to policy transfer in fixed-domain reinforcement learning through the APT-RL algorithm and contributes a viewpoint of advantage-based policy regularization

**Claims And Evidence:**

No

**Claims Explanation:**

**Summary**

After careful consideration of the reviewers’ assessments, I recommend rejecting this submission.

While the paper introduces a novel approach to policy transfer in fixed-domain reinforcement learning through the APT-RL algorithm and contributes a viewpoint of advantage-based policy regularization, the reviewers collectively express significant reservations about the manuscript’s clarity, empirical substantiation, and broader applicability.

**Key Concerns**
- Evidence vs. Claims: Two reviewers (3BJ1 and h9eJ) explicitly flagged misalignment between the experimental claims and the presented evidence. Reviewer 3BJ1 pointed out inconsistencies between the narrative and the plots, which were not addressed despite follow-up inquiries. Reviewer h9eJ also emphasized a perceived lack of attention to detail and rigor in the manuscript's preparation.

- Novelty and Scope Limitations: Reviewer NMr9 appreciated the conceptual simplicity of using the advantage as a regularization term but noted that this idea closely parallels existing methods such as REPAINT. More critically, they raised a concern that APT-RL presupposes strong similarity between source and target policies—an assumption that severely restricts the method's utility in real-world cross-domain transfer learning.

- Responsiveness and Author Engagement: Multiple reviewers expressed concern over the lack of author responsiveness, especially to critical questions and clarifications, which further erodes confidence in the submission’s current form.

**Resubmission Of Major Revision:**

The authors may consider submitting a major revision at a later time.